# Inductive Biases and Variable Creation in Self-Attention Mechanisms

## Abstract

Self-attention, an architectural motif designed to model long-range interactions in sequential data, has driven numerous recent breakthroughs in natural language processing and beyond. This work provides a theoretical analysis of the inductive biases of self-attention modules, where our focus is to rigorously establish which functions and long-range dependencies self-attention blocks prefer to represent. Our main result shows that bounded-norm Transformer layers create sparse variables: they can represent sparse functions of the input sequence, with sample complexity scaling only logarithmically with the context length. Furthermore, we propose new experimental protocols to support this analysis and to guide the practice of training Transformers, built around the large body of work on provably learning sparse Boolean functions.

## 1 Introduction

Self-attention mechanisms have comprised a dramatic paradigm shift in deep learning in recent years, appearing ubiquitously in recent empirical breakthroughs in sequence modeling and unsupervised representation learning. Starting with large-scale natural language processing (Vaswani et al., 2017), self-attention has enjoyed surprising empirical successes in numerous and diverse modalities of data. In many of these settings, self-attention has supplanted traditional recurrent and convolutional architectures, which are understood to incorporate inductive biases about temporal and translational invariances in the data. Self-attention models discard these functional forms, in favor of directly and globally modeling long-range interactions within the input context.

The proliferation of self-attention raises countless mysteries for theorists and empiricists. One fundamental question concerns its statistical properties: *How should we think about the inductive biases of self-attention models?* More specifically, we can ask: *Which functions do self-attention blocks prefer to represent? How many (approximately) distinct functions can they represent?* To this end, this work initiates an analysis of the statistical foundations of self-attention, as it is used in today's state-of-the-art models.

Our main technical contribution is a classical norm-based generalization bound for a Transformer network, which can be extended to related and future architectures via a modular abstraction of attention mechanisms. In particular, the capacity (in terms of log covering number) of the function class of bounded weight self-attention heads (and Transformers) grows only logarithmically in the context length, which provides theoretical justification for the empirical observation that attention models can learn long-term dependencies without overfitting. Next, we show that bounded-norm self-attention heads are capable of representing wide classes of sparse functions. This representational capacity result, combined with the generalization results, provides a partial theoretical explanation for the observed sparsity bias of attention models, which we term *sparse variable creation*.

We accompany this analysis with an experimental study of the sample complexity needed by Transformers to learn sparse Boolean conjunctions, and verify the sample complexity scaling law predicted by the theory in this clean synthetic setting. We discuss how to extend and repurpose this experimental protocol of benchmarking long-context sequence models on synthetic "cryptographic" tasks. We find that Transformers trained with gradient descent can learn sparse parities with noise, which may be of independent interest, exposing the empirical study of Transformers to the rich theory established around this problem.

## 1.1 RELATED WORK

The direct precursors to modern self-attention architectures were recurrent and convolutional networks augmented with attention mechanisms (Bahdanau et al., 2014; Luong et al., 2015; Xu et al., 2015). Landmark work by Vaswani et al. (2017) demonstrated significantly improved machine translation models via self-attention only; autoregressive language models followed shortly (Liu et al., 2018; Radford et al., 2018; 2019; Brown et al., 2020), as well as self-supervised representation learning (Devlin et al., 2018). More recently, self-attention has demonstrated promise in computer vision (Dosovitskiy et al., 2020), protein folding (Jumper et al., 2021), theorem proving (Polu & Sutskever, 2020), program synthesis (Chen et al., 2021b), and reinforcement learning (Chen et al., 2021a; Janner et al., 2021).

**Norm-based generalization bounds.** There is a vast body of literature dedicated to establishing statistical guarantees for neural networks, including VC-dimension and shattering bounds (dating back to Anthony et al. (1999)). In recent years, generalization bounds have been established for various architectures under norm bounds including (Bartlett et al., 2017; Neyshabur et al., 2015; 2017; Golowich et al., 2018; Long & Sedghi, 2019; Chen et al., 2019) using covering-based arguments; Jiang et al. (2019) provide an extensive empirical study of how well these bounds predict generalization in practice. Our work complements these bounds by establishing norm-based bounds for attention models. Our main results rely on a novel reduction to the $\ell_\infty$ covering number bound for linear function classes given by Zhang (2002).

**Theory for attention models.** Our work complements various existing theoretical perspectives on attention-based models. Vuckovic et al. (2020) formulate a dynamical system abstraction of attention layers, arriving at similar Lipschitz constant calculations to ours (which are coarser-grained, since they focus on contractivity and stability rather than finite-sample statistical guarantees). Zhang et al. (2019); Snell et al. (2021) study idealizations of optimization problems for self-attention heads. Wei et al. (2021) propose a definition of *statistically meaningful approximation* of function classes that ties statistical learnability with expressivity, and show that Boolean circuits can be *SM-approximated* by Transformers with a sample complexity bound that depends mildly on circuit depth (rather than context size), using a margin amplification procedure.[1] See Appendix D for a broader survey of related work, including universal function approximation-style results (which ignore statistical considerations).

## 2 BACKGROUND AND NOTATION

**Notation.** We use $d$ to denote the embedding dimension for input tokens. $T$ denotes the number of tokens in an input sequence, a.k.a. the *context length* or *context size* of an attention mechanism. And $m$ refers to the number of samples (input sequences) in a data set. $\|\cdot\|_2$ denotes the spectral norm for matrices, and $\|\cdot\|_{p,q}$ denotes the $(p, q)$ matrix norm where the $p$-norm is over columns and $q$-norm over rows. $\|\cdot\|_p$ denotes the $\ell_p$ norm for vectors. For $\ell_2$-norm, we drop the subscript. $B$ is generally used to quantify bounds on norms of matrices and $L$ for Lipschitz constants. $\Delta^{n-1}$ denotes the simplex of dimension $n$, that is, $\Delta^{n-1} := \{x \in \mathbb{R}^n : x \geq 0, \|x\|_1 = 1\}$.

**Covering numbers and uniform generalization bounds.** Our main technical contribution is a generalization bound arising from carefully counting the number of functions representable by a Transformer. This main complexity notion we use is covering number.

We will use the following definition of $\infty$-norm covering number adapted from Zhang (2002):

**Definition 2.1** (Covering number). *For a given class of vector-valued functions $\mathcal{F}$, the covering number $\mathcal{N}_\infty(\mathcal{F}; \varepsilon; \{z^{(i)}\}_{i=1}^m; \|\cdot\|)$ is the smallest size of a collection (a cover) $\mathcal{C} \subset \mathcal{F}$ such that $\forall f \in \mathcal{F}, \exists \widehat{f} \in \mathcal{C}$ satisfying $\max_i \|f(z^{(i)}) - \widehat{f}(z^{(i)})\| \leq \varepsilon$. Further, define $\mathcal{N}_\infty(\mathcal{F}, \varepsilon, m, \|\cdot\|) = \sup_{z^{(1)}\ldots z^{(m)}} \mathcal{N}_\infty(\mathcal{F}; \varepsilon; z^{(1)}, \ldots, z^{(m)}, \|\cdot\|)$.*

If $\mathcal{F}$ is real-valued (instead of vector-valued), we drop the norm from the notation. Furthermore for functions parameterized by a set of parameters $\Theta$, we exploit the notation to replace $\mathcal{F}$ by $\Theta$.

---

[1] Quoting the discussion following Theorem 4.1 of (Wei et al., 2021): *"The correct norm-based Rademacher complexity bound to use for Transformers is unclear."*

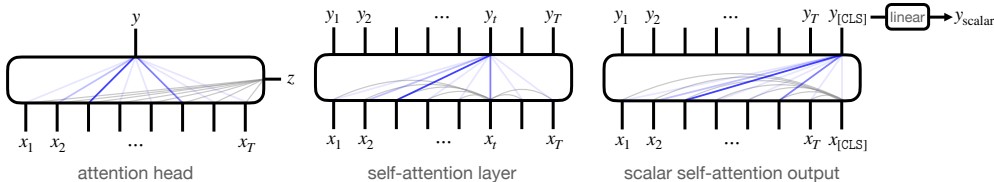

Figure 1: Diagrams of attention modules described in Section 3: alignment scores (grey edges) determine normalized attention weights (blue), which are used to mix the inputs $x_{1:T}$. *Left:* Attention with a general context. *Center:* Stackable self-attention layer, where $x_{1:T}$ is the input as well as the context. *Right:* Auxiliary `[CLS]` token to extract a single scalar from a self-attention layer, providing a real-valued function class for classification or regression tasks.

Recall from Zhang (2002) that for the class of linear functions, $\mathcal{F}_{\text{lin}} = \{x \mapsto w \cdot x : w \in \mathbb{R}^d, \|w\|_s \leq B_W\}$, we have the covering number bound of $\mathcal{N}_\infty(\mathcal{F}; \varepsilon; \{x^{(i)}\}_{i=1}^m) \leq O\left(B_X^2 B_W^2/\varepsilon^2 \cdot \log(B_X B_W m/\varepsilon)\right)$, where $\|x^{(i)}\| \leq B_X$ for $i \in [m]$. Importantly, note that the covering number has a mild dependence on $m$, only logarithmic; this logarithmic dependence on $m$ will be helpful when we turn our analysis to the capacity of attention mechanisms.

**Generalization bounds.** This work focuses on providing log-covering number bounds, which determine the generalization error via standard Rademacher complexity and chaining arguments. The following lemma relates these quantities; we refer the reader to Appendix A.1 for more details.

**Lemma 2.2** (Generalization bound via covering number). *Let $\mathcal{D}$ be a distribution over $\mathcal{X} \times \mathbb{R}$ and let $\ell : \mathbb{R} \times \mathbb{R}$ be a b-bounded loss function that is L-Lipschitz in its first argument. For a given function class $\mathcal{F}$ and $f \in \mathcal{F}$, let $\textsf{risk}(f; \mathcal{D}) := \mathbb{E}_{(x,y) \sim \mathcal{D}}[\ell(f(x), y)]$ and $\widehat{\textsf{risk}}\left(f; (z^{(i)}, y^{(i)})_{i=1}^m\right) := \frac{1}{m} \sum_{i=1}^m \ell(f(z^{(i)}), y^{(i)})$. Suppose $\mathcal{F}$ satisfies $|f| \leq A$ for all $f \in \mathcal{F}$ and $\log \mathcal{N}_\infty(\mathcal{F}; \varepsilon; x^{(1)}, \ldots, x^{(m)}) \leq C_\mathcal{F}/\varepsilon^2$ for all for all $x^{(1)}, \ldots, x^{(m)} \in \mathcal{X}^m$. Then for any $\delta > 0$, with probability at least $1 - \delta$, simultaneously for all $f \in \mathcal{F}$ and some constant $c > 0$,*

$$\left|\textsf{risk}(f; \mathcal{D}) - \widehat{\textsf{risk}}\left(f; (x^{(i)}, y^{(i)})_{i=1}^m\right)\right| \leq 4cL\sqrt{\frac{C_\mathcal{F}}{m}}\left(1 + \log\left(A\sqrt{m/C_\mathcal{F}}\right)\right) + 2b\sqrt{\frac{\log(1/\delta)}{2m}}.$$

## 3 ABSTRACTIONS OF ATTENTION AND SELF-ATTENTION

Attention is not straightforward to define — unlike other architectural components such as convolutions and residual connections, it is a broader model design principle, with numerous variations possible. In this section, we present an abstraction of attention mechanisms, guided by the different manifestations discussed in (Luong et al., 2015), with the goal of enabling a more unified and modular statistical analysis. Subsequently, we show how to represent the Transformer (the predominant attention-based architecture) as a special case of this formulation. Our goal is not necessarily an all encompassing formulation of attention mechanisms, but rather an abstraction that helps to guide general design principles and theoretical analysis.

### 3.1 ATTENTION

Intuitively, we would like to capture the notion that an output variable selects ("attends to") a part of the input context on which it will depend, based on a learned function of global interactions (see Figure 1, left). To this end, we define an *attention head* as a function which maps a context $X$ of $T$ inputs $\{x_t \in \mathcal{X}\}_{t=1}^T$ (e.g. the tokens in a sentence, pixels in an image, or intermediate activations in a neural network) and a context $z \in \mathcal{Z}$ to an output $y \in \mathcal{Y}$. In this work, we will exclusively consider $\mathcal{X}, \mathcal{Y}, \mathcal{Z}$ to be $\mathbb{R}^d$; $d$ is the *embedding dimension*, and we define the matrix of inputs $X = [x_1 x_2 \ldots x_T]^\top \in \mathbb{R}^{T \times d}$. An attention head uses $z$ to select the inputs in $X$ to which the output $y$ will "attend", which we formalize below:

**Definition 3.1** (Attention head). *An attention head is a function $f : \mathcal{X} \to \mathcal{Y}$, specified by an "alignment score" function* $\textsf{Score} : \mathcal{X} \times \mathcal{Z} \to \mathbb{R}$ *parameterized by $\theta_s \in \Theta_s$, normalization function* $\textsf{Norm} : \mathbb{R}^T \to \Delta^{T-1}$, *and position-wise transformations $\phi_{\text{in}} : \mathcal{X} \to \mathcal{V}, \phi_{\text{out}} : \mathcal{V} \to \mathcal{Y}$ parameter-*

*ized by $\theta_{\text{in}} \in \Theta_{\text{in}}$ and $\theta_{\text{out}} \in \Theta_{\text{out}}$. The output of an attention head on input $X \in \mathcal{X}^T, z \in \mathcal{Z}$ is*

$$y = \phi_{\text{out}}\left(\sum_{t=1}^{T}\left[\mathsf{Norm}\Big(\mathsf{Score}(x_1, z; \theta_s), \ldots, \mathsf{Score}(x_T, z; \theta_s)\Big)\right]_t \phi_{\text{in}}(x_t; \theta_{\text{in}}); \ \theta_{\text{out}}\right)$$

$$= \phi_{\text{out}}\left(\phi_{\text{in}}(X; \theta_{\text{in}})^{\top}\mathsf{Norm}\Big(\mathsf{Score}(x_1, z; \theta_s), \ldots, \mathsf{Score}(x_T, z; \theta_s)\Big); \ \theta_{\text{out}}\right)$$

*where $\phi_{\text{in}}(X; \theta) = [\phi_{\text{in}}(x_1; \theta) \ldots \phi_{\text{in}}(x_T; \theta)]^{\top}$ denotes the row-wise application of $\phi_{\text{in}}$.*

The above definition corresponds to leftmost diagram in Figure 1. Here, $\mathcal{V}$ is a vector space of input representations "mixed" by the normalized alignment scores; in this work, we will set $\mathcal{V} = \mathbb{R}^k$. A function class of attention heads is induced by specifying parameter classes for $\{\Theta_s, \Theta_{\text{in}}, \Theta_{\text{out}}\}$.

## 3.2 Self-attention and Transformers

A *self-attention head* is a special case of an attention head, in which the context $z$ is one of the inputs $x_t$ themselves: interactions between elements in $X$ are used to select the elements of $X$ on which $f$ depends. In this case, we will use the term *context* to denote $X$. The focus of this work is to analyze the inductive biases of such a construction. For example, for a self-attention head (see Figure 1 (center)), we would have that the $t$-th component is:

$$y_t = \phi_{\text{out}}\left(\phi_{\text{in}}(X; \theta_{\text{in}})^{\top}\mathsf{Norm}(\mathsf{Score}(X, x_t; \theta_s)); \theta_{\text{out}}\right),$$

We now define the Transformer self-attention architecture as a special case of the above. Since a Transformer layer has shared parameters between multiple output heads, it will be convenient to define all $T$ outputs of this layer at once.

**Definition 3.2** (Transformer self-attention layer). *A Transformer attention layer is a collection of $T$ attention heads with outputs $y_1, \ldots, y_T$, specified by the following choices of function classes (with shared parameters between the heads) where the context for $y_\tau$ is $x_\tau$.*

- $\mathsf{Score}(x, x_\tau; \{W_Q, W_K\}) := x_\tau^{\top} W_Q W_K^{\top} x, \quad W_Q, W_K \in \mathbb{R}^{d \times k}$*(for output $y_t$)*
- $\phi_{\text{in}}(x; W_V) := W_V^{\top} x, \quad W_V \in \mathbb{R}^{d \times k}$
- $\phi_{\text{out}}(x; W_C) := W_C^{\top} \sigma(x), \quad W_C \in \mathbb{R}^{k \times d}$, $L_\sigma$-*Lipschitz function $\sigma : \mathbb{R} \to \mathbb{R}$ applied position-wise, with $\sigma(0) = 0$.*
- $\mathsf{Norm}(x) := \mathsf{softmax}(x) = \frac{\exp(x)}{\mathbf{1}^{\top}\exp(x)}$

*Defining $Y := [y_1 y_2 \ldots y_T]^{\top} \in \mathbb{R}^{T \times d}$ and $[\mathsf{RowSoftmax}(M)]_{t,:} := \mathsf{softmax}(M_{t,:})$, we have*

$$Y = \sigma\left(\mathsf{RowSoftmax}\left(XW_Q(XW_K)^{\top}\right)XW_V\right)W_C.$$

Functions from the above class of Transformer layers map $\mathbb{R}^{T \times d}$ to itself, so that instances from this function class can be composed. Although Definition 3.2 only contains the "self-attention" component, it is not merely a simplified idealization of the full Transformer architecture used in practice. We discuss some remaining discrepancies (positional embeddings, layer normalization, parallel heads, position-wise feedforward networks) in Section 4.3 and the appendix.

**Extracting scalar outputs from a Transformer.** We introduce one more construction: the canonical way to extract a scalar prediction from the final layer of a Transformer. This is the setup used by the classification modules in BERT (Devlin et al., 2018) and all of its derivatives. For a context of size $T$, a Transformer layer with $T + 1$ inputs is constructed, with a special input index `[CLS]`. The input at this position is a vector $x_{\texttt{[CLS]}} \in \mathbb{R}^d$ (which can be considered as a constant, a part of the input, or a trainable parameter); the output is a linear function $w^{\top} y_{\texttt{[CLS]}}$, for a trainable parameter $w \in \mathbb{R}^d$. This defines a class of functions mapping $\mathbb{R}^{T \times d} \to \mathbb{R}$, parameterized by a Transformer layer's parameters and $w$, which we call the class of *scalar-output Transformers*.

## 4 Capacity measures of attention modules

In this section, we present our main technical results, along with overviews of their proofs. Section 4.1 bounds the capacity of a general attention head. Section 4.2 instantiates this bound for the

case of a single Transformer self-attention head. Section 4.3 generalizes this bound for full depth-$L$ Transformer networks. Our sample complexity guarantees scale only *logarithmically* in the context length $T$, providing rigorous grounding for the intuition that the architecture's inductive bias selects sparse functions of the context. Lastly, in Section 4.4, we complement this capacity analysis by exhibiting classes of functions expressible using low-norm Transformer architectures. Combining these representation results and corresponding capacity bounds, we coin the term *sparse variable creation* to refer to this inductive bias.

**Note:** Throughout this section, assume that $\|x_t\|_2 \leq B_X$ for all $t \in [T]$. Note that this allows for the Frobenius norm $\|X\|_F$ to scale with $\sqrt{T}$. The key challenge throughout our analysis is to avoid incurring factors of norms which take a sum over the $t$ dimension, by analyzing the attention parameters in appropriately chosen geometries.

### 4.1 CAPACITY OF A GENERAL ATTENTION HEAD

Recall that the attention head architecture can be represented as a function $f_{\text{head}} : \mathbb{R}^{T \times d} \times \mathbb{R}^d \to \mathbb{R}^d$ parameterized by $\theta_s, \theta_{\text{in}}, \theta_{\text{out}}$ as

$$f_{\text{head}}(X, z; \theta_s, \theta_{\text{in}}, \theta_{\text{out}}) = \phi_{\text{out}}\left(\phi_{\text{in}}(X; \theta_{\text{in}})^\top \text{Norm}(\text{Score}(X, z; \theta_s)); \theta_{\text{out}}\right).$$

Denote the corresponding function class by $\mathcal{F}_{\text{head}} := \{(X, z) \mapsto f_{\text{head}}(X, z; \theta_s, \theta_{\text{in}}, \theta_{\text{out}}) : \theta_s \in \Theta_s, \theta_{\text{in}} \in \Theta_{\text{in}}, \theta_{\text{out}} \in \Theta_{\text{out}}\}$. To convert the vector-valued function class to a scalar output function class, we define $\mathcal{F}_{\text{scalar}} := \{(X, z) \mapsto w^\top f(X, z) : f \in \mathcal{F}_{\text{head}}, w \in \mathbb{R}^d, \|w\| \leq B_w\}$.

For simplicity, we will focus only on the attention part and assume that $\phi_{\text{out}}$ is a fixed function (no parameters) and $w$ is fixed. It is not hard to handle these even if allowed to be trainable. For the case of Transformers, we handle this more generally (see Appendix A.6).

**Assumption 4.1.** *We make the following assumptions:*
1. *$\phi_{\text{out}}$ is $L_{\text{out}}$-Lipschitz in the $\ell_2$-norm, that is, $\forall a, b \in \mathbb{R}^k, \|\phi_{\text{out}}(a) - \phi_{\text{out}}(b)\| \leq L_{\text{out}}\|a - b\|$.*
2. *$\phi_{\text{in}}$ is $B_{\text{in}}$-bounded in $\ell_2$-norm, that is, $\|\phi_{\text{in}}(a; \theta_{\text{in}})\| \leq B_{\text{in}}\|a\|$ for all $a \in \mathbb{R}^d$ and $\theta_{\text{in}} \in \Theta_{\text{in}}$.*
3. *Norm is continuously differentiable and its Jacobian satisfies $\forall \theta \in \mathbb{R}^T, \|J\,\text{Norm}(\theta)\|_{1,1} \leq C_{\text{Norm}}$.*

The Jacobian assumption might seem strong. However, softmax (the most commonly used Norm function) satisfies this with $C_{\text{softmax}} = 2$ (see Corollary A.7).

We prove the following bound on the covering number of $\mathcal{F}_{\text{head}}$ for $m$ samples,

**Theorem 4.2** (Attention head capacity). *Under Assumptions 4.1, the covering number of $\mathcal{F}_{\text{head}}$ satisfies*

$$\log \mathcal{N}_\infty\left(\mathcal{F}_{\text{head}}; \varepsilon; \left\{(X^{(i)}, z^{(i)})\right\}_{i=1}^m, \|\cdot\|_2\right)$$

$$\leq \inf_{\alpha \in [0,1]} \left[\log \mathcal{N}_\infty\left(\mathcal{F}_{\text{Score}}; \frac{\alpha\varepsilon}{C_{\text{Norm}} L_{\text{out}} B_{\text{in}} B_X}; \{(x_t^{(i)}, z^{(i)})\}_{i \in [m], t \in [T]}\right)\right.$$

$$\left. + \log \mathcal{N}_\infty\left(\mathcal{F}_{\text{in}}; \frac{(1-\alpha)\varepsilon}{L_{\text{out}}}; \{x_t^{(i)}\}_{i \in [m], t \in [T]}; \|\cdot\|_2\right)\right],$$

*where $\mathcal{F}_{\text{Score}} = \{(x, z) \mapsto \text{Score}(x, z; \theta_s) : \theta_s \in \Theta_s\}$, and $\mathcal{F}_{\text{in}} = \{x \mapsto \phi_{\text{in}}(x; \theta_{\text{in}}) : \theta_{\text{in}} \in \Theta_{\text{in}}\}$.*

Note that the bound is in terms of the $\mathcal{N}_\infty$ covering number of functions that dependent on dimensions $d$ or $k$ and not $T$. The effect of $T$ only shows up in the number of samples to cover. The $\mathcal{N}_\infty$ number for many classes scales only logarithmically with the number of samples (for eg, linear functions Zhang (2002)). This is exactly what allows us to get a $\log T$ dependence for Transformers.

Since $w$ is fixed, an $\varepsilon$-covering of $\mathcal{F}_{\text{head}}$ directly gives us an $\varepsilon B_w$-covering for $\mathcal{F}_{\text{scalar}}$ implying,

$$\log \mathcal{N}_\infty\left(\mathcal{F}_{\text{scalar}}; \varepsilon; \left\{(X^{(i)}, z^{(i)})\right\}_{i=1}^m\right) \leq \log \mathcal{N}_\infty\left(\mathcal{F}_{\text{head}}; \varepsilon/B_w; \left\{(X^{(i)}, z^{(i)})\right\}_{i=1}^m, \|\cdot\|_2\right).$$

**Proof overview.** In order to prove the bound, we first show a Lipschitzness property of $f_{\text{tf-head}}$. This property allows us to construct the cover by using the covers for $\mathcal{F}_{\text{Score}}$ and $\mathcal{F}_{\text{in}}$.

**Lemma 4.3** ($\ell_\infty$-Lipschitzness of $f_{\text{tf-head}}$)**.** *For any* $\theta_s, \widehat{\theta}_s \in \Theta_s, \theta_{\text{in}}, \widehat{\theta}_{\text{in}} \in \Theta_{\text{in}}$*; for all* $X \in \mathbb{R}^{T \times d}$*, such that* $\left\| X^\top \right\|_{2,\infty} \leq B_X$,

$$\left\| f_{\text{head}}(X, z; \theta_s, \theta_{\text{in}}, w) - f_{\text{head}}(X, z; \widehat{\theta}_s, \widehat{\theta}_{\text{in}}, w) \right\|$$

$$\leq C_{\text{Norm}} L_{\text{out}} B_{\text{in}} B_X \left\| \text{Score}(X, z; \theta_s) - \text{Score}(X, z; \widehat{\theta}_s) \right\|_\infty + L_{\text{out}} \left\| \phi_{\text{in}}(X; \theta_{\text{in}}) - \phi_{\text{in}}(X; \widehat{\theta}_{\text{in}}) \right\|_{2,\infty}.$$

The most crucial part of this proof is to ensure that we do not get a spurious $T$ dependence when accounting for the attention mechanism. The key observation here is that the attention part of the network is computed using Norm, whose Jacobian norm is bounded. This allows us to use the mean-value theorem to move to the maximum ($\ell_\infty$) error over $T$ tokens instead of sum ($\ell_1$), which could potentially incur a $T$ factor. Furthermore, this allows us to combine all samples and tokens and construct a $\ell_\infty$-cover for $mT$ samples.

## 4.2 CAPACITY OF A TRANSFORMER SELF-ATTENTION HEAD

Let us now look at the case of a Transformer self-attention head and instantiate the covering bound. For ease of presentation and to focus on the self-attention part, we collapse $W_Q W_K^\top$ to a single matrix (this does not change the representation), set $k = d$ and remove the linear layer $W_C$[2]. Then the Transformer self-attention head (for any fixed $\tau$) can be described as

$$f_{\text{tf-head}}(X; W_V, W_{QK}) := \sigma\left( W_V^\top X^\top \text{softmax}\left( X W_{QK}^\top x_\tau \right) \right)$$

which is obtained from the general formulation by setting the context to be $x_\tau$, $\text{Score}(X, x_\tau; W_{QK}) = X W_{QK}^\top x_\tau$, $\text{Norm} = \text{softmax}$ and $\phi_{\text{out}} = \sigma$.

Let us define the function class of self-attention heads with bounded norm, $\mathcal{F}_{\text{tf-head}} := \{X \mapsto f_{\text{tf-head}}(X; W_V, W_{QK}) : \|W_V^T\|_{2,\infty} \leq B_V^\infty, \|W_V\| \leq B_V, \|W_{QK}\|_{2,\infty} \leq B_{QK}^\infty \}$. Since $W_V, W_{QK}$ have dimensions dependent on $d$ and $k$, bounding their norms does not hide a $T$ dependence. As before, to convert this vector-valued function class to a scalar output function class, we define $\mathcal{F}_{\text{tf-scalar}} := \{X \mapsto w^\top f(X) : f \in \mathcal{F}_{\text{tf-head}}, w \in \mathbb{R}^d, \|w\| \leq B_w \}$.

We obtain the following bound on the covering number of $\mathcal{F}_{\text{tf-head}}$ as a corollary of Theorem 4.2:

**Corollary 4.4.** *For any* $\varepsilon > 0$ *and* $X^{(1)}, \ldots, X^{(m)} \in \mathbb{R}^{T \times d}$ *such that* $\left\| X^{(i)^\top} \right\|_{2,\infty} \leq B_X$ *for all* $i \in [m]$, *the covering number of* $\mathcal{F}_{\text{tf-head}}$ *satisfies*

$$\log \mathcal{N}_\infty(\mathcal{F}_{\text{tf-head}}; \varepsilon; X^{(1)}, \ldots, X^{(m)}, \|\cdot\|_2) \lesssim (dL_\sigma B_X)^2 \cdot \frac{\left( (B_V^\infty)^{\frac{2}{3}} + (B_{QK}^\infty B_V)^{\frac{2}{3}} \right)^3}{\varepsilon^2} \cdot \log(mT)$$

*Here* $\lesssim$ *hides logarithmic dependencies on quantities besides* $m$ *and* $T$.

Our bounds have a logarithmic dependence on $T$, highlighting the inductive bias of the transformer towards selecting sparse functions of the context.

**Proof overview.** The above result follows from bounding the covering numbers of $\mathcal{F}_{QK} := \{z \mapsto x_\tau^\top W_{QK} z : \|W_{QK}\|_{2,\infty} \leq B_{QK}^\infty \}$ and $\mathcal{F}_V := \{z \to W_V^\top z : \|W_V^T\|_{2,\infty} \leq B_V^\infty, \|W_V\| \leq B_V \}$.

Note that $|x_\tau^\top W_{QK} x - x_\tau^\top W_{QK} x| \leq \|W_{QK} x - W_{QK} x\|$ since $\|x_\tau\| \leq 1$, so the covering number of $\mathcal{F}_{QK}$ is at most the covering number of the class of functions of the form $x \mapsto W_{QK} x$. Therefore, a bound on the vector-valued linear function class suffices to handle both covering numbers. We derive the following covering bound which gives the desired result.

**Lemma 4.5.** *Let* $\mathcal{W} : \{W \in \mathbb{R}^{d_1 \times d_2} : \|W\|_{2,\infty} \leq B_\infty \}$, *and consider the function class* $\mathcal{F} : \{x \mapsto Wx : W \in \mathcal{W} \}$. *For any* $\varepsilon > 0$ *and* $x^{(1)}, \ldots, x^{(N)} \in \mathbb{R}^{d_1}$ *satisfying* $\forall i \in [N], \|x^{(i)}\| \leq B_X$,

$$\log \mathcal{N}_\infty(\mathcal{F}; \varepsilon; x^{(1)}, \ldots, x^{(N)}; \|\cdot\|_2) \lesssim \frac{(d_2 B_\infty B_X)^2}{\varepsilon^2} \cdot \log(N).$$

---

[2]See Appendix 4.3 for a general analysis.

The proof of Lemma 4.5 actually proves a somewhat stronger bound for the function class given by $\{W \in \mathbb{R}^{d_1 \times d_2} : \|W\|_{2,\infty} \le B_\infty, \|W\|_{2,1} \le B_1\}$, with $d_2 B_\infty B_1$ in the numerator instead of $(d_2 B_\infty)^2$, but we have kept the latter formulation for simplicity of presentation.

Finally, we discuss how to account for some important architectural modifications.

**Positional embeddings.** In practice, the permutation-invariant symmetry of a Transformer network is broken by adding a *positional embedding* matrix $P \in \mathbb{R}^{T \times d}$ to the input $X$ at the first layer. In practice, the embedding matrix is often fixed and non-trainable. Our results extend to this setting in a straightforward way; see Appendix A.4. If these matrices are to be trained from a sufficiently large class (say, $\|P^\top\|_{2,\infty} \le 1$), the dependence of the log-covering number on $T$ could become linear.

**Multi-head self-attention.** In almost all applications of Transformers, multiple parallel self-attention heads are used, and their outputs aggregated, to allow for a richer representation. Our analysis directly extends to this setting; see Appendix A.5 for details. When a single attention head is replaced with the sum of $H$ parallel heads, the log-covering number scales up by a factor of $\mathrm{poly}(H)$.

**Layer normalization.** State-of-the-art Transformer networks are trained with layer normalization modules (Ba et al., 2016), which is generally understood to aid optimization. We keep a variant of layer normalization in the covering number analysis– it proves to be useful in the analysis of full attention blocks (see Appendix A.6), as it keeps the norm of the embedding of each token bounded. Removing these layers would lead to a worse dependence on the spectral norm of the matrices.

### 4.3 CAPACITY OF DEEP TRANSFORMER NETWORKS

In this section, we will extend our results for $L$-layer Transformer blocks. Denote the weights of layer $i$ by $W^{(i)} := \left\{ W_Q^{(i)}, W_K^{(i)}, W_V^{(i)}, W_C^{(i)} \right\}$. Further denote the set of weights up to layer $i$ by $W^{1:i} = (W^{(1)}, \ldots, W^{i-1})$. Denote the input representation of layer $i$ by $g_{\text{tf-block}}^{(i)}(X; W^{1:i})$. We inductively define $g_{\text{tf-block}}^{(i)} : \mathbb{R}^{T \times d} \to \mathbb{R}^{T \times d}$ starting with $g_{\text{tf-block}}^{(1)}(X; W^{1:1}) = X$ (the input):

$$g_{\text{tf-block}}^{(i+1)}\left(X; W^{1:i+1}\right) := \Pi_{\text{norm}}\left(\sigma\left(\Pi_{\text{norm}}\left(f\left(g_{\text{tf-block}}^{(i)}\left(X; W^{1:i}\right); W^{(i)}\right)\right)\right) W_C^{(i)}\right)$$

$$\text{with } f\left(Z; \{W_Q, W_K, W_V, \cdot\}\right) := \mathsf{RowSoftmax}\left(ZW_Q \left(ZW_K\right)^\top\right) ZW_V,$$

where $\Pi_{\text{norm}}$ denotes layer normalization[3] applied to each row. We use a slightly modified version of LayerNorm where instead of normalizing to norm 1, we project it to the unit ball. Let us denote the class of depth-$L$ transformer blocks by

$$\mathcal{F}_{\text{tf-block}}^{(L)} := \left\{ X \to g_{\text{tf-block}}^{(L+1)}(X; W^{1:L+1}) : \forall i \in [L], \left\|W_V^{(i)}\right\|_2, \left\|W_K^{(i)} W_Q^{(i)\top}\right\|_2, \left\|W_C^{(i)}\right\|_2 \le C_2, \right.$$

$$\left. \left\|W_V^{(i)}\right\|_{2,\infty}, \left\|W_K^{(i)} W_Q^{(i)\top}\right\|_{2,\infty}, \left\|W_C^{(i)}\right\|_{2,\infty} \le C_\infty \right\}.$$

To obtain a final scalar output, we use a linear function of the [CLS] output, $g_{\text{tf-scalar}}(X; W^{1:L+1}, w) = w^\top \left[g\left(X; W^{1:L+1}\right)\right]_{[\text{CLS}],:}$. Let the scalar output function class be $\mathcal{F}_{\text{tf-scalar}}^{(L)} := \{X \to w^\top f(X)_{[\text{CLS}]} : f \in \mathcal{F}_{\text{tf-block}}^{(L)}, w \in \mathbb{R}^d, \|w\| \le B_w\}$.

**Theorem 4.6** (Theorem A.17 (Simplified)). *Suppose $\forall i \in [m], \|X^{(i)}\|_{2,\infty} \le B_X$, then we have*

$$\log \mathcal{N}_\infty(\mathcal{F}_{\text{tf-block}}^{(L)}; \varepsilon; X^{(1)}, \ldots, X^{(m)}) \lesssim (C_2 L_\sigma)^{O(L)} \cdot \frac{d^2 B_X^2 B_w^2 C_\infty^2}{\varepsilon^2} \cdot \log(mT).$$

Note that the dependence on $T$ is only logarithmic even for deeper networks. The dependence on embedding dimension and $(2, \infty)$ norms of the weight matrices is quadratic. As long as the spectral norms of the matrices are bounded by 1 and $\sigma$ is 1-Lipschitz (which holds for sigmoids and ReLUs), the exponential dependence on $L$ can be avoided.

---

[3]Layer normalization allows for the norms of the outputs of each token in each layer to remain bounded by 1. Note that the norm of the entire input can still have a dependence on $T$. Our results would go through with a worse dependence on the spectral norms if we were to remove layer norm.

### 4.4 SPARSE VARIABLE CREATION: AN INDUCTIVE BIAS FOR SELF-ATTENTION

The above analysis shows that function classes bottlenecked by self-attention mechanisms are "small" in terms of the context size. In this section, we answer the converse question: *which functions of interest can they express?* To this end, we show in this section that sparse Boolean functions are realizable by bounded-norm Transformers.

Given a Boolean function $f : \{0, 1\}^T \to \mathbb{R}$ which only depends on $s$ of its inputs, we represent $f$ using a self-attention head $f_{\text{tf-head}}$ composed with a feedforward network $f_{\text{mlp}}$; this is the repeated block in the standard Transformer architecture. Intuitively, $f_{\text{tf-head}}$ can select an $s$-dimensional subset of inputs to "attend to", while $f_{\text{mlp}}$ memorizes an arbitrary function of these $s$ inputs (requiring up to $\approx 2^s$ parameters). In the regime of $s \ll \log T$ (think of $f_{\text{tf-head}} \circ f_{\text{mlp}}$ as implementing a single composable Boolean gate), the corresponding statistical guarantees are meaningful. In order to describe this combination of sample-efficient sparsification of rich contexts and subsequent restricted use of universal function approximation, we coin the term *sparse variable creation*.

**Setup.** We consider the classes of Boolean functions $f : \{0, 1\}^T \to \mathbb{R}$ representable by bounded-norm scalar-output Transformer heads $f_{\text{tf-scalar}} : \mathbb{R}^{T \times d} \to \mathbb{R}$. To do this, we must first fix a mapping from $\{0, 1\}^T$ to $\mathbb{R}^{T \times d}$; we discuss several natural choices in Appendix B.1. The simplest of these uses a sum of token and positional embeddings $X(b)_{t,:} := e_{b_t} + v_t$, for a set of approximately orthogonal unit vectors $\{e_0, e_1, v_1, \ldots, v_T\}$. After choosing a mapping $X(b)$, the setup of the representation problem is evident: given $f(b)$, find Transformer weights $\theta_{\text{tf-head}}$ and feedforward network weights $\theta_{\text{mlp}}$ such that

$$f_{\text{tf+mlp}}(X(b); \theta_{\text{tf-head}}, \theta_{\text{mlp}}) := f_{\text{mlp}}\left(f_{\text{tf-head}}(X(b); \theta_{\text{tf-head}}); \theta_{\text{mlp}}\right) \approx f(b), \qquad \forall b \in \{0, 1\}^T.$$

**Main representational results.** We show that Transformer blocks can represent $\mathcal{I}$-sparse Boolean functions, whose values only depend on some subset of indices $\mathcal{I} \subseteq [T]$. We present informal statements of these approximation results below, and present the precise statements in Appendix B.2.

**Proposition 4.7** (Sparse variable creation via Transformers; informal). *Under any of the input mappings $X(b)$, we have the following guarantees:*

- *$f_{\text{tf-scalar}}$ alone can approximate a particular monotone symmetric $s$-sparse Boolean function, with weight norms $\|W_Q\|_F \leq O\left(\log(Ts)\right)$, $\|W_K\|_F$, $\|W_V\|_F$, $\|W_C\|_F \leq O(s)$.*
- *$f_{\text{tf+mlp}}$ can exactly represent symmetric $s$-sparse functions, with the same Transformer weight norms as above; the feedforward network weights satisfy $\|W_1\|_F$, $\|W_2\|_F$, $\|w\|_F \leq O(\text{poly}(s))$.*
- *$f_{\text{tf+mlp}}$ can exactly represent general $s$-sparse functions, with the same Transformer weight norms as above; the feedforward network weights satisfy $\|W_1\|_F$, $\|W_2\|_F$, $\|w\|_F \leq O(2^s \cdot \text{poly}(s))$.*

**Proof ideas.** Each construction uses the same idea: select $W_Q, W_K$ so that the attention mixture weights approximate the uniform distribution over the relevant positions, then use the ReLU network to memorize all distinct values of $f$. Full proofs are given in Appendix B.4.

## 5 EMPIRICAL SCALING LAWS FOR LEARNING BOOLEAN GATES

Our theoretical analysis has shown that Transformers can represent sparse Boolean functions, with sample complexity scaling mildly with the context size. In this section, we present an empirical study of whether Transformer architectures (as they are trained and used in state-of-the-art language modeling) exhibit these scalings in practice.

We introduce a rigorous benchmark for probing the empirical sample complexity of a Transformer: attribute-efficient learning of a planted sparse Boolean function. We choose a family of distinct distributions $\{\mathcal{D}_1, \ldots, \mathcal{D}_N\}$ on $\{0, 1\}^T \times \{0, 1\}$, corresponding to supervised learning problems, such that the feature distributions are identical, and the uniform mixture $\frac{1}{N} \sum_{i=1}^N \mathcal{D}_i$ is invariant under all permutations of the indices $1, \ldots, T$. We then select an $i^* \in [N]$ uniformly at random, train a Transformer binary classifier on $m$ samples from $\mathcal{D}_{i^*}$, then evaluate the generalization error via cross-validation. Since the architecture, initialization, training algorithm, and training data are permutation-invariant, at least $\Omega(\log N)$ samples are required to learn this distribution: one sample can only reveal a single bit of information about $i^*$, via its binary label. We are interested in the empirical scaling of the sufficient sample size $m$ to solve this problem, in terms of $N$.

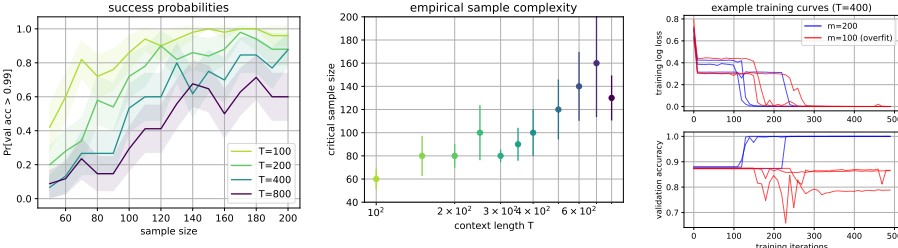

Figure 2: Statistically probing a Transformer by training it on a 3-way AND of a hidden subset of i.i.d. random bits. *Left, center:* Sublinear scaling of the empirical sample complexity. *Right:* Example training curves in the {overfitting, correct} regimes: $T = 400, m = \{100, 200\}$.

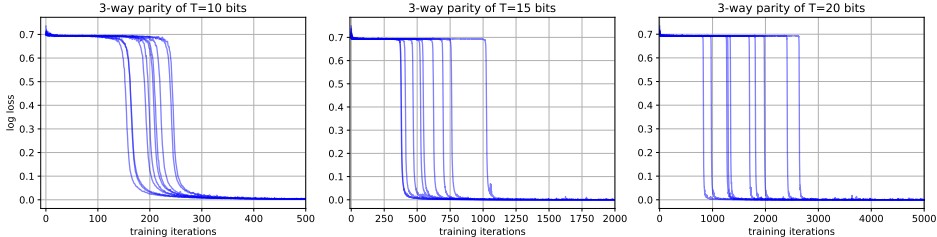

Figure 3: A curious empirical finding: Transformers can learn sparse parities. 10 loss curves (with online batches) are given for this setup with $s = 3, T = \{10, 15, 20\}$, showing abrupt phase transitions from random guessing to perfect classification. See Appendix C.2 for details.

**Learning sparse** AND **gates.** The simplest instantiation of this experimental framework is a hidden $s$-way AND under the uniform distribution (i.e. $T$ i.i.d. random bits). The Transformer must learn which of the $N = \binom{T}{s}$ combinations of inputs determines the label, which requires $m \geq \Omega(s \log T)$ samples. The theory predicts the sample complexity of learning a bounded-norm Transformer should match this scaling in $T$. With hyperparameters typical of Transformer setups used for natural data, we indeed observe that the empirical sample complexity scales sublinearly with $T$. Figure 2 summarizes our findings; details are provided in Appendix C.1.

**Learning sparse parities.** We can also replace the sparse AND operation with XOR: the label is the parity of a hidden subset of input bits. This variant emphasizes the "cryptographic" nature of this experimental setup, due to its known computational hardness. $\Omega(T^s)$ statistical queries (thus, batch gradient descent steps) are necessary (Kearns, 1998); in the presence of noise, the fastest known algorithms for learning parities with noise require $T^{\Omega(s)}$ time (Valiant, 2012). Figure 3 (with details in Appendix C.2) show that Transformer models can fit sparse parities. This raises an intriguing question: if theory suggests that "exhaustive search-like" methods are computationally necessary for this problem, why does local search (i.e. gradient-based training) succeed? We leave this *computational* (as opposed to statistical) mystery as an open direction for future work.

## 6 CONCLUSION AND FUTURE WORK

We have presented a theoretical analysis of the inductive biases of self-attention models, finding that they can learn sparse Boolean functions with sample complexity scaling logarithmically in the context length. We call this phenomenon *sparse variable creation*. Our analysis is accompanied by new empirical probes involving training Transformers on sparse Boolean functions, where we corroborate this scaling of the sample complexity in practice. We believe our capacity bounds are improvable (we have only sought to obtain an optimal dependence on $T$). Incorporating aspects of computation and depth remains a perennial challenge in this line of inquiry.

Building further upon the principles of attention constitutes a vibrant frontier of empirical research (Tolstikhin et al., 2021; Lee-Thorp et al., 2021; Jaegle et al., 2021b;a; d'Ascoli et al., 2021). We hope that the theoretical foundations and experimental probes presented in this paper will assist in further expanding the breadth of applications of self-attention, and developing more compute-efficient, data-efficient, controllable, and reliable algorithmic interventions.

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

# A    PROOFS OF CAPACITY BOUNDS

In this section we present the full proofs (including the omitted proofs) of our capacity bounds. We also cover relevant background and useful technical lemmas.

## A.1    RADEMACHER COMPLEXITY AND GENERALIZATION BOUNDS

Here we briefly review Rademacher complexity and its relationship to covering numbers and generalization bounds. We refer the reader to Bartlett & Mendelson (2002) for a more detailed exposition.

**Definition A.1** (Empirical Rademacher complexity). *For a given class of functions $\mathcal{F} = \{f : \mathcal{X} \to \mathbb{R}\}$ and $\{z^{(i)} \in \mathcal{X}\}_{i=1}^{m}$, the empirical Rademacher complexity $\widehat{\mathcal{R}}(\mathcal{F}; z^{(1)}, \ldots, z^{(m)})$ is defined as*

$$\widehat{\mathcal{R}}(\mathcal{F}; z^{(1)}, \ldots, z^{(m)}) = \frac{1}{m} \mathbb{E}_\varepsilon \left[ \sup_{f \in \mathcal{F}} \sum_{i=1}^{m} \varepsilon_i f(z^{(i)}) \right],$$

*where $\varepsilon$ is a vector of $m$ i.i.d. Rademacher random variables ($\Pr[\varepsilon_i = 1] = \Pr[\varepsilon_i = -1] = 1/2$).*

In order to relate the Rademacher complexity and $\ell_\infty$-covering numbers, we use a modified version of Dudley's metric entropy.

**Lemma A.2** (Dudley (1967); modified). *Consider a real-valued function class $\mathcal{F}$ such that $|f| \le A$ for all $f \in \mathcal{F}$. Then*

$$\widehat{\mathcal{R}}(\mathcal{F}; z^{(1)}, \ldots, z^{(m)}) \le c \cdot \inf_{\delta \ge 0} \left( \delta + \int_\delta^A \sqrt{\frac{\log \mathcal{N}_\infty(\mathcal{F}; \varepsilon; z^{(1)}, \ldots, z^{(m)})}{m}} \, d\varepsilon \right)$$

*for some constant $c > 0$.*

*Proof sketch.* The original statement is for 2-norm covering number, but the $\infty$-norm case reduces to the 2-norm case because $N_2(\cdot) \le N_\infty(\cdot)$. The original statement also fixes $\delta = 0$ rather than taking an infimum. Also, the standard statement has the integral go from 0 to $\infty$, but these are easily replaced with $\delta$ and $A$.    $\square$

For our paper, we will instantiate the above lemma for log covering numbers scaling as $1/\varepsilon^2$.

**Corollary A.3** (Rademacher Complexity via covering number). *Consider a real-valued function class $\mathcal{F}$ such that $|f| \le A$ for all $f \in \mathcal{F}$. Suppose $\log \mathcal{N}_\infty(\mathcal{F}; \varepsilon; z^{(1)}, \ldots, z^{(m)}) \le C_\mathcal{F}/\varepsilon^2$, then*

$$\widehat{\mathcal{R}}(\mathcal{F}; z^{(1)}, \ldots, z^{(m)}) \le c \cdot \sqrt{\frac{C_\mathcal{F}}{m}} \cdot \left( 1 + \log \left( A \sqrt{m/C_\mathcal{F}} \right) \right)$$

*for some constant $c > 0$.*

*Proof.* Using Lemma A.2, we have for some constant $c > 0$,

$$\begin{aligned}
\widehat{\mathcal{R}}(\mathcal{F}; z^{(1)}, \ldots, z^{(m)}) &\le c \inf_{\delta \ge 0} \left( \delta + \int_\delta^A \sqrt{\frac{\log \mathcal{N}_\infty(\mathcal{F}; \varepsilon; z^{(1)}, \ldots, z^{(m)})}{m}} \, d\varepsilon \right) \\
&\le c \inf_{\delta \ge 0} \left( \delta + \int_\delta^A \sqrt{\frac{C_\mathcal{F}}{\varepsilon^2 m}} \, d\varepsilon \right) \\
&= c \inf_{\delta \ge 0} \left( \delta + \sqrt{\frac{C_\mathcal{F}}{m}} \int_\delta^A \frac{1}{\varepsilon} \, d\varepsilon \right) \\
&= c \inf_{\delta \ge 0} \left( \delta + \sqrt{\frac{C_\mathcal{F}}{m}} \log(A/\delta) \right) \\
&= c \sqrt{\frac{C_\mathcal{F}}{m}} \left( 1 + \log \left( A \sqrt{m/C_\mathcal{F}} \right) \right).
\end{aligned}$$

$\square$

We can now obtain a generalization guarantee from the Rademacher complexity of a function class:

**Theorem A.4** (Bartlett & Mendelson (2002)). *Let $\mathcal{D}$ be a distribution over $\mathcal{X} \times \mathbb{R}$ and let $\ell : \mathbb{R} \times \mathbb{R}$ be a b-bounded loss function that is L-Lipschitz in its first argument. For a given function class $\mathcal{F}$ and $f \in \mathcal{F}$, let $\mathrm{risk}(f; \mathcal{D}) := \mathbb{E}_{(x,y) \sim \mathcal{D}}[\ell(f(x), y)]$ and $\widehat{\mathrm{risk}}\left(f; (z^{(i)}, y^{(i)})_{i=1}^m\right) := \frac{1}{m} \sum_{i=1}^m \ell(f(z^{(i)}), y^{(i)})$. Then for any $\delta > 0$, with probability at least $1 - \delta$, simultaneously for all $f \in \mathcal{F}$,*

$$\left| \mathrm{risk}(f; \mathcal{D}) - \widehat{\mathrm{risk}}\left(f; (z^{(i)}, y^{(i)})_{i=1}^m\right) \right| \leq 4L\,\widehat{\mathcal{R}}\left(\mathcal{F}; z^{(1)}, \ldots, z^{(m)}\right) + 2b\sqrt{\frac{\log(1/\delta)}{2m}}.$$

Combining the above, we get:

**Lemma A.5** (Lemma 2.2 (restated)). *Consider a function class $\mathcal{F}$ such that $|f| \leq A$ for all $f \in \mathcal{F}$ and $\log \mathcal{N}_\infty(\mathcal{F}; \varepsilon; x^{(1)}, \ldots, x^{(m)}) \leq C_{\mathcal{F}}/\varepsilon^2$ for all $x^{(1)}, \ldots, x^{(m)} \in \mathcal{X}^m$. Then for any $\delta > 0$, with probability at least $1 - \delta$, simultaneously for all $f \in \mathcal{F}$,*

$$\left| \mathrm{risk}(f; \mathcal{D}) - \widehat{\mathrm{risk}}\left(f; (x^{(i)}, y^{(i)})_{i=1}^m\right) \right| \leq 4cL\sqrt{\frac{C_{\mathcal{F}}}{m}}\left(1 + \log\left(A\sqrt{m/C_{\mathcal{F}}}\right)\right) + 2b\sqrt{\frac{\log(1/\delta)}{2m}},$$

*for some constant $c > 0$.*

## A.2 USEFUL LEMMAS

**Lemma A.6.** *Consider function $f : \mathbb{R}^d \to \Delta^{d-1}$ such that the Jacobian of the function satisfies $\|J\,f(\theta)\|_{1,1} \leq c_f$ for all $\theta \in \mathbb{R}^d$, then for any vectors $\theta_1, \theta_2 \in \mathbb{R}^p$,*

$$\|f(\theta_1) - f(\theta_2)\|_1 \leq c_f \|\theta_1 - \theta_2\|_\infty.$$

*Proof.* By the fundamental theorem of calculus applied to $g(t) = f(t\theta_1 + (1-t)\theta_2)$, followed by a change of variables:

$$f(\theta_1) - f(\theta_2) = \left(\int_0^1 J\left(t\theta_1 + (1-t)\theta_2\right) dt\right)(\theta_1 - \theta_2),$$

We have

$$\|f(\theta_1) - f(\theta_2)\|_1 = \left\|\int_0^1 J\left(t\theta_1 + (1-t)\theta_2\right)(\theta_1 - \theta_2) dt\right\|_1$$

By Jensen's inequality:

$$\leq \int_0^1 \|J\left(t\theta_1 + (1-t)\theta_2\right)(\theta_1 - \theta_2)\|_1\, dt$$

Using $\|Ax\|_1 \leq \|A\|_{1,1}\|x\|_\infty$:

$$\leq \int_0^1 \|J\left(t\theta_1 + (1-t)\theta_2\right)\|_{1,1}\|\theta_1 - \theta_2\|_\infty\, dt$$

By assumption on the Jacobian:

$$\leq c_f \|\theta_1 - \theta_2\|_\infty.$$

$\square$

**Corollary A.7.** *For vectors $\theta_1, \theta_2 \in \mathbb{R}^p$, $\|\mathrm{softmax}(\theta_1) - \mathrm{softmax}(\theta_2)\|_1 \leq 2\|\theta_1 - \theta_2\|_\infty$.*

*Proof.* Observe that for softmax, the Jacobian satisfies:

$$J(\theta) = \mathrm{diag}(\mathrm{softmax}(\theta)) - \mathrm{softmax}(\theta)\mathrm{softmax}(\theta)^\top.$$

We have for all $\theta, h$,

$$\|J(\theta)\|_{1,1} = \sum_{i=1}^p \sum_{j=1}^p |\mathrm{softmax}(\theta)_i(\mathbb{1}[i=j] - \mathrm{softmax}(\theta)_j)|$$

$$= \sum_{i=1}^{p} \mathsf{softmax}(\theta)_i \left( 1 - \mathsf{softmax}(\theta)_i + \sum_{j \neq i} \mathsf{softmax}(\theta)_j \right)$$

$$= 2 \sum_{i=1}^{p} \mathsf{softmax}(\theta)_i \left( 1 - \mathsf{softmax}(\theta)_i \right)$$

$$\leq 2.$$

Combining the above with Lemma A.6 gives the desired result. $\qquad \square$

**Lemma A.8.** *For $\alpha_i, \beta_i \geq 0$, the solution to the following optimization*

$$\min_{x_1,\ldots,x_n} \sum_{i=1}^{n} \frac{\alpha_i}{x_i^2}$$

$$\textit{subject to } \sum_{i=1}^{n} \beta_i x_i = C$$

*is $\frac{\gamma^3}{C^2}$ and is achieved at $x_i = \frac{C}{\gamma} \left( \frac{\alpha_i}{\beta_i} \right)^{1/3}$ where $\gamma = \sum_{i=1}^{n} \alpha_i^{1/3} \beta_i^{\frac{2}{3}}$.*

*Proof.* The proof follows by a standard Lagrangian analysis. $\qquad \square$

**Lemma A.9** (Contraction of $\Pi_{\mathsf{norm}}$). *Let $\Pi_{\mathsf{norm}}$ be the projection operator onto the unit norm ball. For any vectors $u, v$, we have $\|\Pi_{\mathsf{norm}}(u) - \Pi_{\mathsf{norm}}(v)\| \leq \|u - v\|$.*

*Proof.* If $u, v$ are both in the unit ball then this follows trivially. Let us assume that $\|u\| \geq \|v\|$ and $\|u\| \geq 1$ WLOG. First suppose $\|v\| \leq 1$. Let $B_V^{(1)} = \alpha u$ be the projection of $v$ in the direction of $u$, and let $B_V^2 = v - B_V^{(1)}$. Then

$$\begin{aligned} \|\Pi_{\mathsf{norm}}(u) - \Pi_{\mathsf{norm}}(v)\|^2 &= \|u/\|u\| - v\|^2 \\ &= \|u/\|u\| - (\alpha u + B_V^2)\|^2 \\ &= \|(\|u\|^{-1} - \alpha)u - B_V^2\|^2 \\ &= (\|u\|^{-1} - \alpha)^2 \|u\|^2 + \|B_V^2\|^2 \\ &\leq (1 - \alpha^2)\|u\|^2 + \|B_V^2\|^2 \qquad \text{since } \|u\|^{-1} < \alpha < 1 \\ &= \|u - (\alpha u + B_V^2)\|^2 \\ &= \|u - v\|^2 \end{aligned}$$

If $\|v\| > 1$, then

$$\|\Pi_{\mathsf{norm}}(u) - \Pi_{\mathsf{norm}}(v)\| = \|\Pi_{\mathsf{norm}}(u/\|v\|) - \Pi_{\mathsf{norm}}(v/\|v\|)\| \leq \|u/\|v\| - v/\|v\|\| < \|u - v\|.$$

where the second-to-last inequality follows from the $\|v\| < 1$ case. $\qquad \square$

**Lemma A.10** (Zhang (2002) Theorem 4). *Let $\mathcal{V} : \{v : v \in \mathbb{R}^{d_1}, \|v\| \leq B_1\}$ and $\mathcal{F}_{linear} = \{x \mapsto v^\top x : v \in \mathcal{V}\}$. For any $\delta > 0$ and $x^{(1)}, \ldots, x^{(N)}$ satisfying $\|x^{(i)}\| \leq B_2 \, \forall i$,*

$$\log \mathcal{N}_\infty(\mathcal{F}_{linear}; \varepsilon; x^{(1)}, \cdots, x^{(N)}) \leq 36 \frac{B_1^2 B_2^2}{\varepsilon^2} \log(2\lceil 4B_1 B_2/\varepsilon + 2 \rceil N + 1).$$

## A.3 OMITTED PROOFS

*Proof of Lemma 4.3.* Observe that,

$$\left\| f_{\mathsf{head}}(X, z; \theta_s, \theta_{\mathsf{in}}) - f_{\mathsf{head}}(X, z; \widehat{\theta}_s, \widehat{\theta}_{\mathsf{in}}) \right\|$$

$$= \left\| \phi_{\mathsf{out}} \left( \phi_{\mathsf{in}}(X; \theta_{\mathsf{in}})^\top \mathsf{Norm}(\mathsf{Score}(X, z; \theta_s)) \right) - \phi_{\mathsf{out}} \left( \phi_{\mathsf{in}}(X; \widehat{\theta}_{\mathsf{in}})^\top \mathsf{Norm}(\mathsf{Score}(X, z; \widehat{\theta}_s)) \right) \right\|$$

By $L_{\text{out}}$-Lipschitzness of $\phi_{\text{out}}$ and bound on $\|w\|$:

$$\leq L_{\text{out}} \left\| \phi_{\text{in}}(X; \theta_{\text{in}})^\top \mathsf{Norm}(\mathsf{Score}(X, z; \theta_s)) - \phi_{\text{in}}(X; \widehat{\theta}_{\text{in}})^\top \mathsf{Norm}(\mathsf{Score}(X, z; \widehat{\theta}_s)) \right\|$$

By triangle inequality:

$$\leq L_{\text{out}} \left\| \phi_{\text{in}}(X; \theta_{\text{in}})^\top \left( \mathsf{Norm}(\mathsf{Score}(X, z; \theta_s)) - \mathsf{Norm}(\mathsf{Score}(X, z; \widehat{\theta}_s)) \right) \right\|$$
$$+ L_{\text{out}} \left\| \left( \phi_{\text{in}}(X; \theta_{\text{in}}) - \phi_{\text{in}}(X; \widehat{\theta}_{\text{in}}) \right)^\top \mathsf{Norm}(\mathsf{Score}(X, z; \widehat{\theta}_s)) \right\|$$

Using $\|Pv\| \leq \|P\|_{2,\infty} \|v\|_1$ and $B_{\text{in}}$-boundedness of $\phi_{\text{in}}$:

$$\leq L_{\text{out}} B_{\text{in}} \left\| \mathsf{Norm}(\mathsf{Score}(X, z; \theta_s)) - \mathsf{Norm}(\mathsf{Score}(X, z; \widehat{\theta}_s)) \right\|_1$$
$$+ L_{\text{out}} \left\| \left( \phi_{\text{in}}(X; \theta_{\text{in}}) - \phi_{\text{in}}(X; \widehat{\theta}_{\text{in}}) \right)^\top \right\|_{2,\infty} \left\| \mathsf{Norm}(\mathsf{Score}(X, z; \widehat{\theta}_s)) \right\|_1$$

By Lemma A.6 and the assumption on Norm:

$$\leq L_{\text{out}} C_{\mathsf{Norm}} \left\| \phi_{\text{in}}(X; \theta_{\text{in}})^\top \right\|_{2,\infty} \left\| \mathsf{Score}(X, z; \theta_s) - \mathsf{Score}(X, z; \widehat{\theta}_s) \right\|_\infty + L_{\text{out}} \left\| \left( \phi_{\text{in}}(X; \theta_{\text{in}}) - \phi_{\text{in}}(X; \widehat{\theta}_{\text{in}}) \right)^\top \right\|_{2,\infty}$$

By boundedness of $\phi_{\text{in}}$ and $\|X^\top\|_{2,\infty} \leq B_X$:

$$\leq L_{\text{out}} C_{\mathsf{Norm}} B_{\text{in}} B_X \left\| \mathsf{Score}(X, z; \theta_s) - \mathsf{Score}(X, z; \widehat{\theta}_s) \right\|_\infty + L_{\text{out}} \left\| \left( \phi_{\text{in}}(X; \theta_{\text{in}}) - \phi_{\text{in}}(X; \widehat{\theta}_{\text{in}}) \right)^\top \right\|_{2,\infty}.$$

$\square$

*Proof of Theorem 4.2.* Our goal is to show that for every $\varepsilon > 0$, collection of inputs $(X^{(1)}, z^{(1)}), \ldots, (X^{(m)}, z^{(m)})$, there is a cover $\mathcal{C}_{\mathsf{head}}$ such that for all $\theta_s \in \Theta_s, \theta_{\text{in}} \in \Theta_{\text{in}}$, there is some $(\widehat{\theta}_s, \widehat{\theta}_{\text{in}}) \in \mathcal{C}_{\mathsf{head}}$ such that $\max_i \left\| f_{\mathsf{head}}(X^{(i)}, z^{(i)}; \theta_s, \theta_{\text{in}}) - f_{\mathsf{head}}(X^{(i)}, z^{(i)}; \widehat{\theta}_s, \widehat{\theta}_{\text{in}}) \right\| \leq \varepsilon$.

Observe that for all $\theta_s, \widehat{\theta}_s$,

$$\max_{i \in [m]} \|\mathsf{Score}(X^{(i)}, z^{(i)}; \theta_s) - \mathsf{Score}(X^{(i)}, z^{(i)}; \widehat{\theta}_s)\|_\infty = \max_{i \in [m], t \in [T]} \left| \mathsf{Score}(x_t^{(i)}, z^{(i)}; \theta_s) - \mathsf{Score}(x_t^{(i)}, z^{(i)}; \widehat{\theta}_s) \right|.$$

Similarly, for all $\theta_{\text{in}}, \widehat{\theta}_{\text{in}}$,

$$\max_{i \in [m]} \left\| \left( \phi_{\text{in}}(X^{(i)}; \theta_{\text{in}}) - \phi_{\text{in}}(X^{(i)}; \widehat{\theta}_{\text{in}}) \right)^\top \right\|_{2,\infty} = \max_{i \in [m], t \in [T]} \left\| \phi_{\text{in}}(x_t^{(i)}; \theta_{\text{in}}) - \phi_{\text{in}}(x_t^{(i)}; \widehat{\theta}_{\text{in}}) \right\|.$$

This crucially allows us to aggregate over the $i$ and $t$ dimensions together.[4] Therefore, we can consider $\mathcal{N}_\infty$ covers for the above to bound the overall covering number.

Let $\mathcal{C}_{\mathsf{Score}}$ be the $\varepsilon_{\mathsf{Score}}$-cover $(\infty)$ for $\mathcal{F}_{\mathsf{Score}}$ over inputs $\left\{ (x_t^{(i)}, z^{(i)}) \right\}_{i \in [m], t \in [T]}$ of size

$$\mathcal{N}_\infty \left( \mathcal{F}_{\mathsf{Score}}; \varepsilon_{\mathsf{Score}}; \{(x_t^{(i)}, z^{(i)})\}_{i \in [m], t \in [T]} \right).$$

Also, Let $\mathcal{C}_{\text{in}}$ be the $\varepsilon_{\text{in}}$-cover $(\infty)$ for $\mathcal{F}_{\text{in}}$ over inputs $\{x_t^{(i)}\}_{i \in [m], t \in [T]}$ of size

$$\mathcal{N}_\infty \left( \mathcal{F}_{\text{in}}; \varepsilon_{\text{in}}; \{x_t^{(i)}\}_{i \in [m], t \in [T]}; \|\cdot\|_2 \right).$$

We are ready to construct the cover for $\mathcal{F}_{\mathsf{head}}$. Set $\mathcal{C}_{\mathsf{head}} = \{f_{\mathsf{head}}(\cdot; \widehat{\theta}_s, \widehat{\theta}_{\text{in}}))_{i \in [m]} : \widehat{\theta}_s \in \mathcal{C}_{\mathsf{Score}}, \widehat{\theta}_{\text{in}} \in \mathcal{C}_{\text{in}}\}$. Then for any $\theta_s \in \Theta_s, \theta_{\text{in}} \in \Theta_{\text{in}}$, there exists $\widehat{\theta}_s, \widehat{\theta}_{\text{in}} \in \mathcal{C}_{\mathsf{head}}$, such that for all $i \in [m]$, using Lemma 4.3:

$$\left\| f_{\mathsf{head}}(X^{(i)}, z^{(i)}; \theta_s, \theta_{\text{in}}) - f_{\mathsf{head}}(X^{(i)}, z^{(i)}; \widehat{\theta}_s, \widehat{\theta}_{\text{in}}) \right\| \leq C_{\mathsf{Norm}} L_{\text{out}} B_{\text{in}} B_X \varepsilon_{\mathsf{Score}} + L_{\text{out}} \varepsilon_{\text{in}}.$$

---

[4]In the case of the Transformer self-attention mechanism, we will obtain $\infty$-norm covering numbers for Score and $\phi_{\text{in}}$ that have only logarithmic dependence on the number of examples. Because of this aggregation trick, the resulting covering number for the whole layer will have merely logarithmic dependence on the context length $T$.

The size of the cover we have constructed is,

$$\log |\mathcal{C}_{\text{head}}| = \log |\mathcal{C}_{\text{Score}}| + \log |\mathcal{C}_{\text{in}}|$$
$$= \log \mathcal{N}_\infty \left( \mathcal{F}_{\text{Score}}; \varepsilon_{\text{Score}}; \{(x_t^{(i)}, z^{(i)})\}_{i \in [m], t \in [T]} \right) + \log \mathcal{N}_\infty \left( \mathcal{F}_{\text{in}}; \varepsilon_{\text{in}}; \{x_t^{(i)}\}_{i \in [m], t \in [T]}; \| \cdot \|_2 \right)$$

and we are done. $\qquad\square$

*Proof of Corollary 4.4.* By Theorem 4.2, the covering number of $\mathcal{F}_{\text{tf-head}}$ satisfies

$$\log \mathcal{N}_\infty \left( \mathcal{F}_{\text{tf-head}}; \varepsilon; \left\{ (X^{(i)}, z^{(i)}) \right\}_{i=1}^m \right)$$
$$\leq \inf_{\alpha \in [0,1]} \left[ \log \mathcal{N}_\infty \left( \mathcal{F}_{QK}; \frac{\alpha \varepsilon}{2 L_\sigma B_V B_X}; \{(x_t^{(i)}, z^{(i)})\}_{i \in [m], t \in [T]} \right) \right.$$
$$\left. + \log \mathcal{N}_\infty \left( \mathcal{F}_V; \frac{(1-\alpha)\varepsilon}{L_\sigma}; \{x_t^{(i)}\}_{i \in [m], t \in [T]}; \| \cdot \|_2 \right) \right].$$

where we have used the fact that for a scalar-output Transformer layer:

- softmax satisfies the Jacobian assumption with $C_{\text{softmax}} = 2$ using Corollary A.7.
- $L_{\text{out}}$ is the Lipschitz constant of $\sigma$: $L_\sigma$.
- $B_{\text{in}}$ is a bound on the norm of $W_V^\top x$ with respect to norm of $x$: $B_V$.

By Lemma 4.5, for any $\varepsilon_{QK}, \varepsilon_V > 0$:

$$\log \mathcal{N}_\infty \left( \mathcal{F}_{QK}; \varepsilon_{QK}; \{(x_t^{(i)}, z^{(i)})\}_{i \in [m], t \in [T]} \right) \lesssim \frac{(d B_{2,\inf}^{QK} B_X)^2 \log(mT)}{\varepsilon_{QK}^2}$$

$$\log \mathcal{N}_\infty \left( \mathcal{F}_V; \varepsilon_V; \{(x_t^{(i)}, z^{(i)})\}_{i \in [m], t \in [T]}; \| \cdot \|_2 \right) \lesssim \frac{(d B_{2,\inf}^V B_X)^2 \log(mT)}{\varepsilon_V^2}$$

since $W_{QK}, W_V \in \mathbb{R}^{d \times d}$ ($k = d$). We want to choose $\varepsilon_{QK}$ and $\varepsilon_V$ to minimize the sum of the above two terms, subject to

$$2 L_\sigma B_V B_X \varepsilon_{QK} + L_\sigma \varepsilon_V \leq \varepsilon.$$

By Lemma A.8, the solution to this optimization leads to an optimal bound of:

$$\log \mathcal{N}_\infty(\mathcal{F}_{\text{tf-head}}; \varepsilon; X^{(1)}, \ldots, X^{(M)}) \lesssim (d L_\sigma B_X)^2 \cdot \frac{\left( (B_V^\infty)^{\frac{2}{3}} + (B_{QK}^\infty B_V B_X)^{\frac{2}{3}} \right)^3}{\varepsilon^2} \cdot \log(mT).$$

$\qquad\square$

*Proof of Lemma 4.5.* Let $B_\infty$ be an upper bound on $\|W\|_{2,\infty}$ and $B_1$ be an upper bound on $\|W\|_{2,1}$.

The approach will be to cover each of the columns of $W$ independently, treating each as specifying a linear function from $\mathbb{R}^{d_1} \to \mathbb{R}$.

By Lemma A.10, letting $\mathcal{V}(b) : \{v : v \in \mathbb{R}^{d_1}, \|v\| \leq b\}$ and $\mathcal{F}_{\text{linear}}(b) = \{x \mapsto v^\top x : v \in \mathcal{V}(b)\}$, for any $\delta > 0$

$$\log \mathcal{N}_\infty(\mathcal{F}_{\text{linear}}(b); \delta; x^{(1)}, \cdots, x^{(N)}) \leq \frac{c b^2 B_X^2 \log((1 + b B_X / \delta) N)}{\delta^2}.$$

given that $\|x^{(i)}\| \leq B_X$ for all $i$.

In fact the cover, which we denote by $\widehat{\mathcal{F}}_{\text{linear}}(b; \delta)$, is proper: $\widehat{\mathcal{F}}_{\text{linear}}(b; \delta) = \{x \mapsto \widehat{v}^\top x : \widehat{v} \in \widehat{V}\}$ for some finite subset $\widehat{V} \subset \mathcal{V}(b)$.

Let

$$\mathcal{S} = \left\{ (k_1, k_2, \ldots, k_{d_2}) : k_i \in \{0, 1, \ldots, d_2\} \text{ for all } i \text{ and } 0 \leq \sum_{i=1}^{d_2} k_i \leq d_2 \right\}$$

Given any $W \in \mathcal{W}$, let $v_i$ denote the $i$th column of $W$. For each $i$, let

$$k_i^* = \left\lfloor \frac{d_2}{B_\infty} \|v_i\| \right\rfloor$$

Let $S_W = (k_1^*, \ldots, k_{d_2}^*)$. Then $S_W \in \mathcal{S}$, and

$$\frac{B_\infty}{d_2} k_i^* \leq \|v_i\| \leq \frac{B_\infty}{d_2}(k_i^* + 1) \quad \forall i.$$

For every tuple $S = (k_1, k_2, \ldots, k_{d_2}) \in \mathcal{S}$, let

$$\widehat{\mathcal{W}}_S = \left\{ [\widehat{v}_1 \widehat{v}_2 \ldots \widehat{v}_{d_2}]^\top : \widehat{v}_i \in \widehat{\mathcal{F}}_{\text{linear}} \left( B_\infty(k_i + 1)/d_2; \varepsilon\sqrt{(k_i + 1)/(2d_2)} \right) \text{ for all } i \right\}.$$

Our cover will be $\widehat{\mathcal{F}} : \{x \mapsto \widehat{W}x : \widehat{W} \in \bigcup_{S \in \mathcal{S}} \widehat{\mathcal{W}}_S\}$. Note that

$$|\widehat{\mathcal{F}}| \leq \sum_{S \in \mathcal{S}} \prod_{i=1}^{d_2} \mathcal{N}_\infty \left( \mathcal{F}_{\text{linear}}(B_\infty(k_i + 1)/d_2); \varepsilon\sqrt{(k_i + 1)/(2d_2)}; x^{(1)}, \cdots, x^{(N)} \right)$$

$$\lesssim \sum_{S \in \mathcal{S}} \exp \left( \sum_{i=1}^{d_2} \frac{B_\infty^2 a^2 (k_i + 1)^2 \log(N)}{d_2^2 \varepsilon^2 (k_i + 1)/(2d_2)} \right)$$

$$= \sum_{S \in \mathcal{S}} \exp \left( \frac{2B_\infty^2 B_X^2 \log(N)}{d_2 \varepsilon^2} \sum_{i=1}^{d_2} (k_i + 1) \right)$$

$$\leq |\mathcal{S}| \exp \left( \frac{4B_1 B_\infty B_X^2 \log(N)}{\varepsilon^2} \right)$$

where in the last step we used the fact that

$$\sum_{i=1}^{d_2} k_i \leq \frac{d_2}{B_\infty} \sum_{i=1}^{d} \|v_i\| = d_2 B_1 / B_\infty$$

Since $|\mathcal{S}| \leq \binom{2d_2}{d_2} = \exp(O(d_2 \log d_2))$, we obtain

$$\log |\widehat{\mathcal{F}}| \lesssim \frac{d_2 B_1 B_\infty B_X^2 \log(N)}{\varepsilon^2}$$

as desired. In particular, we obtain the more concise, but looser, bound from the lemma statement by using the fact that $B_1 \leq d_2 B_\infty$

For a particular $W = [v_1 \ldots v_{d_2}] \in \mathcal{W}$ it is guaranteed that there is a matrix $\widehat{W} \in \widehat{\mathcal{W}}_{S_W}$, $\widehat{W} = [\widehat{v}_1 \ldots \widehat{v}_{d_2}]$, such that for each $i \in [d_2]$,

$$|v_i^\top x_n - \widehat{v}_i^\top x_n| \leq \varepsilon\sqrt{\frac{k_i + 1}{2d_2}} \quad \forall n \in [N].$$

where $k_i$ is the $i$th element of $S_W$. We then obtain the desired covering property:

$$\max_{n \in [N]} \left\| W x_n - \widehat{W} x_n \right\| = \max_{n \in [N]} \sqrt{\sum_{i=1}^{d_2} (v_i^\top x_n - \widehat{v}_i^\top x_n)^2}$$

$$\leq \sqrt{\sum_{i=1}^{d_2} \varepsilon^2 \left( \frac{k_i + 1}{2d_2} \right)}$$

$$= \varepsilon \sqrt{\frac{d_2 + \sum_{i=1}^{d_2} k_i}{2d_2}} \quad \leq \varepsilon$$

$\square$

### A.4 Capacity with positional embeddings

Since the Transformer architecture is permutation invariant for all $t \neq \tau$, positional embeddings (fixed or trainable) are typically added to the inputs to distinguish the different positions of the tokens. These positional embeddings are matrices $P \in \mathbb{R}^{T \times d}$ such that $P = [p_1 \ldots p_T]^\top$ for $p_i \in \mathbb{R}^d$. Accounting for the positional embeddings as input, a single Transformer attention head can be expressed as:

$$f_{\text{tf-pos}}(X, P; W_V, W_{QK}) := \sigma \left( W_V^\top (X + P)^\top \mathsf{softmax} \left( (X + P) W_{QK}^\top (x_\tau + p_\tau) \right) \right).$$

For a fixed positional embedding $P$, let us define

$$\mathcal{F}_{\text{tf-pos}}(P) := \{ X \to f_{\text{tf-pos}}(X, P; W_V, W_{QK}) : \|W_V^\top\|_{2,\infty} \leq B_V^\infty, \|W_V\| \leq B_V, \|W_{QK}\|_{2,\infty} \leq B_{QK}^\infty \}$$

. Position embedding just impacts the input into the covering bound argument which effects the bound in terms of the $\left\| P^\top \right\|_{2,\infty}$ as given below,

**Lemma A.11.** *For all $X^{(1)}, \ldots, X^{(m)} \in \mathbb{R}^{T \times d}$ such that $\left\| X^{(i)^\top} \right\|_{2,\infty} \leq B_X$ for all $i \in [m]$, and $P \in \mathbb{R}^{T \times d}$ such that $\|P^\top\|_{2,\infty} \leq B_P$, the covering number of $\mathcal{F}_{\text{tf-pos}}(P)$ satisfies*

$$\log \mathcal{N}_\infty(\mathcal{F}_{\text{tf-pos}}(P); \varepsilon; X^{(1)}, \ldots, X^{(m)}, \|\cdot\|_2) \lesssim (dL_\sigma(B_X + B_P))^2 \cdot \frac{\left( (B_V^\infty)^{\frac{2}{3}} + (2B_{QK}^\infty B_V (B_X + B_P))^{\frac{2}{3}} \right)^3}{\varepsilon^2} \cdot \log(mT).$$

*Proof.* Observe that $f_{\text{tf-pos}}(X, P; W_V, W_{QK}) = f_{\text{tf-head}}(X + P; W_V, W_{QK})$. Thus we have,

$$\log \mathcal{N}_\infty \left( \mathcal{F}_{\text{tf-pos}}(P); \varepsilon; \left\{ (X^{(i)}) \right\}_{i=1}^m, \|\cdot\|_2 \right) = \log \mathcal{N}_\infty \left( \mathcal{F}_{\text{tf-head}}; \varepsilon; \left\{ X^{(i)} + P \right\}_{i=1}^m, \|\cdot\|_2 \right).$$

For all $i \in [m]$, $\left\| (X^{(i)} + P)^\top \right\|_{2,\infty} \leq \left\| X^{(i)^\top} \right\|_{2,\infty} + \left\| P^\top \right\|_{2,\infty} \leq B_X + B_P$. Therefore, using Corollary 4.4, we get the desired result. $\qquad \square$

Therefore our bounds go through for fixed positional embeddings. If we were to train the embeddings, we would need a much finer cover on the embeddings which could incur a $T$ dependence.

### A.5 Capacity of multiple parallel heads

In virtually all practical applications of Transformers since their inception, instead of using one set of weights for an attention head, there are parallel attention heads, which have separate identically-shaped parameters; their outputs are concatenated. For the purposes of this analysis, suppose we have

$$f_{\text{tf-heads}} \left( X; \left\{ W_V^{[h]}, W_{QK}^{[h]} \right\}_{h=1}^H \right) := \sum_{h=1}^H f_{\text{tf-head}} \left( X; W_V^{[h]}, W_{QK}^{[h]} \right).$$

Let us define the class of multi-head self-attention with $H$ heads as

$$\mathcal{F}_{\text{tf-heads}} := \left\{ X \mapsto f_{\text{tf-heads}} \left( X; \left\{ W_V^{[h]}, W_{QK}^{[h]} \right\}_{h=1}^H \right) : \right.$$
$$\left. \forall h \in [H], \|W_V^{[h]^\top}\|_{2,\infty} \leq B_V^{\infty[h]}, \|W_V^{[h]}\| \leq B_V^{[h]}, \|W_{QK}^{[h]}\|_{2,\infty} \leq B_{QK}^{\infty[h]} \right\}.$$

**Lemma A.12.** *For all $X^{(1)}, \ldots, X^{(m)} \in \mathbb{R}^{T \times d}$ such that $\left\| X^{(i)^\top} \right\|_{2,\infty} \leq B_X$ for all $i \in [m]$, the covering number of $\mathcal{F}_{\text{tf-heads}}$ satisfies*

$$\log \mathcal{N}_\infty(\mathcal{F}_{\text{tf-heads}}; \varepsilon; X^{(1)}, \ldots, X^{(m)}, \|\cdot\|_2) \lesssim (dL_\sigma B_X)^2 \cdot \frac{\left( \sum_{h=1}^H (B_V^{\infty[h]})^{\frac{2}{3}} + (2B_{QK}^{\infty[h]} B_V^{[h]})^{\frac{2}{3}} \right)^3}{\varepsilon^2} \cdot \log(mT).$$

*Proof.* For all $h \in [H]$, let $\mathcal{C}_h$ be an $\varepsilon_h$-covering of $\mathcal{F}_{\text{tf-head}}$ with weight bounds corresponding to head $h$. Since $f_{\text{tf-heads}} \left( X; \left\{ W_V^{[h]}, W_{QK}^{[h]} \right\}_{h=1}^H \right) = \sum_{h=1}^H f_{\text{tf-head}} \left( X; W_V^{[h]}, W_{QK}^{[h]} \right)$, we have

$\mathcal{C} := \mathcal{C}_1 \times \ldots \times \mathcal{C}_H{}^5$ is an $\left(\sum_{h=1}^{H} \varepsilon_h\right)$-covering for $\mathcal{F}_{\text{tf-heads}}$. Using Corollary 4.4 (and optimizing for $\varepsilon_h$ using Lemma A.8, by breaking them into individual errors for each head), we have

$$\log|\mathcal{C}| = \sum_{h=1}^{H} \log|\mathcal{C}_h| \leq \sum_{h=1}^{H} \leq (dL_\sigma B_X)^2 \cdot \frac{\left(\sum_{h=1}^{H}(B_V^{\infty\,[h]})^{\frac{2}{3}} + (2B_{QK}^{\infty\,[h]}B_V^{[h]})^{\frac{2}{3}}\right)^3}{\varepsilon^2} \cdot \log(mT).$$

$\square$

To see the dependence on $H$, consider the setting where the weight bounds are the same for each head (dropping the $[h]$ subscript), then we get,

$$\log\mathcal{N}_\infty(\mathcal{F}_{\text{tf-heads}}; \varepsilon; X^{(1)}, \ldots, X^{(m)}, \|\cdot\|_2) \lesssim (dL_\sigma B_X)^2 \cdot H^3 \cdot \frac{\left((B_V^\infty)^{\frac{2}{3}} + (2B_{QK}^\infty B_V)^{\frac{2}{3}}\right)^3}{\varepsilon^2} \cdot \log(mT).$$

## A.6 CAPACITY OF DEEP TRANSFORMER NETWORKS

We will consider an $L$-layer transformer. Let us denote the weights of layer $i$ by $W^{(i)} := \left\{W_Q^{(i)}, W_K^{(i)}, W_V^{(i)}, W_C^{(i)}\right\}$ such that $\left\|W_K^{(i)}W_Q^{(i)\top}\right\|_2 \leq B_{QK}^{(i)}, \left\|W_V^{(i)}\right\|_2 \leq B_V^{(i)}, \left\|W_C^{(i)}\right\|_2 \leq B_C^{(i)}$ and $\left\|W_K^{(i)}W_Q^{(i)\top}\right\|_{2,\infty} \leq B_{QK}^{\infty\,(i)}, \left\|W_V^{(i)}\right\|_{2,\infty} \leq B_V^{\infty\,(i)}$ and $\left\|W_C^{(i)}\right\|_{2,\infty} \leq B_C^{\infty\,(i)}$. Let us further denote the set of weights up to layer $i$ by $W^{1:i} = (W^{(1)}, \ldots, W^{i-1})$. Let the input representation of layer $i$ be $g_{\text{tf-head}}^{(i)}(X; W^{1:i})$. We inductively define $g$ with $g_{\text{tf-head}}^{(1)}(X; W^{1:1}) = X$

$$g_{\text{tf-head}}^{(i+1)}\left(X; W^{1:i+1}\right) = \Pi_{\text{norm}}\left(\sigma\left(\Pi_{\text{norm}}\left(f\left(g_{\text{tf-head}}^{(i)}\left(X; W^{1:i}\right); W^{(i)}\right)\right)\right) W_C^{(i)}\right) \text{ with}$$

$$f\left(Z; \{W_Q, W_K, W_V, W_C\}\right) = \mathsf{RowSoftmax}\left(ZW_Q\left(ZW_K\right)^\top\right) ZW_V,$$

where $\Pi_{\text{norm}}$ is applied row-wise. Our final output is $g_{\text{tf-scalar}}(X; W^{1:L+1}, w) = w^\top g_{\text{tf-head}}^{(L)}\left(X; W^{1:L+1}\right)$ [CLS] for $\|w\| \leq B_w$.

In order to construct a cover, we will first bound the distance between the function $g$ with different weight parameters $W^{1:L+1}$ and $\widehat{W}^{1:L+1}$. This bound will depend on the closeness of the parameters which will allow us to construct a cover of the network in an iterative fashion by constructing covers of each layer.

### A.6.1 LIPSCHITZNESS OF THE NETWORK

To bound the Lipschitzness of the network, we will first bound the distance between $f$ with different weights and inputs.

**Lemma A.13** (Instantiation of Lemma 4.3). *For any $W_K, \widehat{W}_K, W_V, \widehat{W}_V, W_Q, \widehat{W}_Q \in \mathbb{R}^{d \times k}$, for all $Z \in \mathbb{R}^{T \times d}$ such that $\left\|Z^\top\right\|_{2,\infty} \leq 1$,*

$$\left\|\left(f\left(Z; \{W_Q, W_K, W_V, \cdot\}\right) - f\left(Z; \{\widehat{W}_Q, \widehat{W}_K, \widehat{W}_V, \cdot\}\right)\right)^\top\right\|_{2,\infty}$$

$$\leq 2\|W_V\|_2 \left\|\left(W_Q W_K^\top - \widehat{W}_Q \widehat{W}_K^\top\right) Z^\top\right\|_{2,\infty} + \left\|(W_V - \widehat{W}_V)^\top Z^\top\right\|_{2,\infty}$$

*Proof.* Consider a fixed row $\tau$ of the output of the functions,

$$\left\|f\left(Z; \{W_Q, W_K, W_V, \cdot\}\right)[\tau] - f\left(Z; \{\widehat{W}_Q, \widehat{W}_K, \widehat{W}_V, \cdot\}\right)[\tau]\right\|$$

$$= \left\|W_V^\top Z^\top \mathsf{softmax}\left(ZW_K W_Q^\top z_\tau\right) - \widehat{W}_V^\top Z^\top \mathsf{softmax}\left(Z\widehat{W}_K \widehat{W}_Q^\top z_\tau\right)\right\|$$

---

[5]Here, $\times$ denotes the Cartesian product: the functions obtained by using the every combination of parameters of each individual cover.

By triangle inequality:

$$\leq \left\| W_V^\top Z^\top \left( \mathsf{softmax}\left(ZW_K W_Q^\top z_\tau\right) - \mathsf{softmax}\left(Z\widehat{W}_K \widehat{W}_Q^\top z_\tau\right)\right)\right\|$$
$$+ \left\|(W_V - \widehat{W}_V)^\top Z^\top \mathsf{softmax}\left(Z\widehat{W}_K \widehat{W}_Q^\top z_\tau\right)\right\|$$

Using $\|Pv\| \leq \|P\|_{2,\infty} \|v\|_1$:

$$\leq \left\| W_V^\top Z^\top \right\|_{2,\infty} \left\|\mathsf{softmax}\left(ZW_K W_Q^\top z_\tau\right) - \mathsf{softmax}\left(Z\widehat{W}_K \widehat{W}_Q^\top z_\tau\right)\right\|_1$$
$$+ \left\|(W_V - \widehat{W}_V)^\top Z^\top \right\|_{2,\infty} \left\|\mathsf{softmax}\left(Z\widehat{W}_K \widehat{W}_Q^\top z_\tau\right)\right\|_1$$

By Corollary A.7, $\left\|Z^\top\right\|_{2,\infty} \leq 1$, $\|PQ\|_{2,\infty} \leq \|P\|_2 \|Q\|_{2,\infty}$, and $\|P^\top\|_2 = \|P\|_2$:

$$\leq 2\|W_V\|_2 \left\|ZW_K W_Q^\top z_\tau - Z\widehat{W}_K \widehat{W}_Q^\top z_\tau\right\|_\infty + \left\|(W_V - \widehat{W}_V)^\top Z^\top\right\|_{2,\infty}$$
$$\leq 2\|W_V\|_2 \left\|\left(W_Q W_K^\top - \widehat{W}_Q \widehat{W}_K^\top\right) Z^\top\right\|_{2,\infty} + \left\|(W_V - \widehat{W}_V)^\top Z^\top\right\|_{2,\infty}.$$

$\square$

**Lemma A.14.** *For any $W_K, W_V, W_Q \in \mathbb{R}^{d\times k}$, for all $Z, \widehat{Z} \in \mathbb{R}^{T\times d}$ such that $\left\|Z^\top\right\|_{2,\infty} \leq 1, \|\widehat{Z}^\top\|_{2,\infty} \leq 1$,*

$$\left\|\left(f\left(Z; \{W_Q, W_K, W_V, \cdot\}\right) - f\left(\widehat{Z}; \{W_Q, W_K, W_V, \cdot\}\right)\right)^\top\right\|_{2,\infty}$$
$$\leq \|W_V\|_2 \left(1 + 4\left\|W_K W_Q^\top\right\|_2\right) \left\|(Z - \widehat{Z})^\top\right\|_{2,\infty}.$$

*Proof.* Consider a fixed row $\tau$ of the output of the functions,

$$\left\|f\left(Z; \{W_Q, W_K, W_V, \cdot\}\right)[\tau] - f\left(\widehat{Z}; \{W_Q, W_K, W_V, \cdot\}\right)[\tau]\right\|$$
$$= \left\|W_V^\top Z^\top \mathsf{softmax}\left(ZW_K W_Q^\top z_\tau\right) - W_V^\top \widehat{Z}^\top \mathsf{softmax}\left(\widehat{Z}W_K W_Q^\top \widehat{z}_\tau\right)\right\|$$

By triangle inequality:

$$\leq \left\|W_V^\top \left(Z - \widehat{Z}\right)^\top \mathsf{softmax}\left(ZW_K W_Q^\top z_\tau\right)\right\| + \left\|W_V^\top \widehat{Z}^\top \left(\mathsf{softmax}\left(ZW_K W_Q^\top z_\tau\right) - \mathsf{softmax}\left(\widehat{Z}W_K W_Q^\top \widehat{z}_\tau\right)\right)\right\|$$

Using $\|Pv\| \leq \|P\|_{2,\infty}\|v\|_1$:

$$\leq \left\|W_V^\top \left(Z - \widehat{Z}\right)\right\|_{2,\infty} \left\|\mathsf{softmax}\left(ZW_K W_Q^\top z_\tau\right)\right\|_1$$
$$+ \left\|W_V^\top \widehat{Z}^\top\right\|_{2,\infty} \left\|\mathsf{softmax}\left(ZW_K W_Q^\top z_\tau\right) - \mathsf{softmax}\left(\widehat{Z}W_K W_Q^\top \widehat{z}_\tau\right)\right\|_1$$

By Corollary A.7, $\left\|\widehat{Z}^\top\right\|_{2,\infty} \leq 1$ and $\|PQ\|_{2,\infty} \leq \|P\|_2 \|Q\|_{2,\infty}$:

$$\leq \|W_V\|_2 \left\|(Z - \widehat{Z})^\top\right\|_{2,\infty} + 2\|W_V\|_2 \left\|ZW_K W_Q^\top z_\tau - \widehat{Z}W_K W_Q^\top \widehat{z}_\tau\right\|_\infty$$

By triangle inequality:

$$\leq \|W_V\|_2 \left\|(Z - \widehat{Z})^\top\right\|_{2,\infty} + 2\|W_V\|_2 \left(\left\|(Z - \widehat{Z})W_K W_Q^\top z_\tau\right\|_\infty + \left\|\widehat{Z}W_K W_Q^\top (z_\tau - \widehat{z}_\tau)\right\|_\infty\right)$$

Since $\left\|\widehat{Z}^\top\right\|_{2,\infty} \leq 1$ and $\|Pv\|_\infty \leq \|P^\top\|_{2,\infty}\|v\|$:

$$\leq \|W_V\|_2 \left(1 + 4\left\|W_K W_Q^\top\right\|_2\right) \left\|(Z - \widehat{Z})^\top\right\|_{2,\infty}.$$

$\square$

With the above lemmas, we are ready to prove the effect of change of weights on $g$.

**Lemma A.15.** *For any $W_1^{i+1}, \widehat{W}_1^{i+1}$ satisfying the norm constraints,*

$$\left\| \left( g_{\text{tf-block}}^{(i+1)}(X; W^{1:i+1}) - g_{\text{tf-block}}^{(i+1)}(X; \widehat{W}^{1:i+1}) \right)^\top \right\|_{2,\infty}$$

$$\leq \left\| \left( W_C^{(i)} - \widehat{W}_C^{(i)} \right)^\top \sigma \left( \Pi_{\text{norm}} \left( f \left( \left( X; \widehat{W}^{1:i} \right); \widehat{W}^{(i)} \right) \right) \right)^\top \right\|_{2,\infty}$$

$$+ L_\sigma B_C^{(i)} B_V^{(i)} \left( 1 + 4 B_{QK}^{(i)} \right) \left\| \left( g_{\text{tf-block}}^{(i)}\left(X; W^{1:i}\right) - g_{\text{tf-block}}^{(i)}\left(X; \widehat{W}^{1:i}\right) \right)^\top \right\|_{2,\infty}$$

$$+ 2 L_\sigma B_C^{(i)} B_V^{(i)} \left\| \left( W_Q^{(i)} W_K^{(i)\top} - \widehat{W}_Q^{(i)} \widehat{W}_K^{(i)\top} \right) g_{\text{tf-block}}^{(i)}\left(X; \widehat{W}^{1:i}\right)^\top \right\|_{2,\infty}$$

$$+ L_\sigma B_C^{(i)} \left\| (W_V - \widehat{W}_V)^\top g_{\text{tf-block}}^{(i)}\left(X; \widehat{W}^{1:i}\right)^\top \right\|_{2,\infty}.$$

*Proof.* Unrolling one layer, we have

$$\left\| \left( g_{\text{tf-head}}^{(i+1)}\left(X; W^{1:i+1}\right) - g_{\text{tf-head}}^{(i+1)}\left(X; \widehat{W}^{1:i+1}\right) \right)^\top \right\|_{2,\infty}$$

$$= \left\| \left( \Pi_{\text{norm}} \left( \sigma \left( \Pi_{\text{norm}} \left( f \left( g_{\text{tf-head}}^{(i)}\left(X; W^{1:i}\right); W^{(i)} \right) \right) \right) W_C^{(i)} \right) \right. \right.$$

$$\left. \left. - \Pi_{\text{norm}} \left( \sigma \left( \Pi_{\text{norm}} \left( f \left( g_{\text{tf-head}}^{(i)}\left(X; \widehat{W}^{1:i}\right); \widehat{W}^{(i)} \right) \right) \right) \widehat{W}_C^{(i)} \right) \right)^\top \right\|_{2,\infty}$$

Using Lemma A.9 for each row:

$$\leq \left\| W_C^{(i)\top} \sigma \left( \Pi_{\text{norm}} \left( f \left( g_{\text{tf-head}}^{(i)}\left(X; W^{1:i}\right); W^{(i)} \right) \right) \right)^\top - \widehat{W}_C^{(i)\top} \sigma \left( \Pi_{\text{norm}} \left( f \left( g_{\text{tf-head}}^{(i)}\left(X; \widehat{W}^{1:i}\right); \widehat{W}^{(i)} \right) \right) \right) \right\|_{2,\infty}$$

By triangle inequality for each row:

$$\leq \underbrace{\left\| W_C^{(i)\top} \left( \sigma \left( \Pi_{\text{norm}} \left( f \left( g_{\text{tf-head}}^{(i)}\left(X; W^{1:i}\right); W^{(i)} \right) \right) \right) - \sigma \left( \Pi_{\text{norm}} \left( f \left( g_{\text{tf-head}}^{(i)}\left(X; \widehat{W}^{1:i}\right); \widehat{W}^{(i)} \right) \right) \right) \right)^\top \right\|_{2,\infty}}_{(A)}$$

$$+ \left\| \left( W_C^{(i)} - \widehat{W}_C^{(i)} \right)^\top \sigma \left( \Pi_{\text{norm}} \left( f \left( g_{\text{tf-head}}^{(i)}\left(X; \widehat{W}^{1:i}\right); \widehat{W}^{(i)} \right) \right) \right)^\top \right\|_{2,\infty}.$$

Let us focus on term $(A)$.

Bounding the norm per row:

$$(A) \leq \left\| W_C^{(i)} \right\|_2 \left\| \sigma \left( \Pi_{\text{norm}} \left( f \left( g_{\text{tf-head}}^{(i)}\left(X; W^{1:i}\right); W^{(i)} \right) \right) \right)^\top - \sigma \left( \Pi_{\text{norm}} \left( f \left( g_{\text{tf-head}}^{(i)}\left(X; \widehat{W}^{1:i}\right); \widehat{W}^{(i)} \right) \right) \right)^\top \right\|_{2,\infty}$$

Since $\sigma$ is $L_\sigma$-Lipschitz and $\left\| W_C^{(i)} \right\|_2 \leq B_C^{(i)}$, for each row:

$$\leq L_\sigma B_C^{(i)} \left\| \Pi_{\text{norm}} \left( f \left( g_{\text{tf-head}}^{(i)}\left(X; W^{1:i}\right); W^{(i)} \right) \right)^\top - \Pi_{\text{norm}} \left( f \left( g \left(X; \widehat{W}^{1:i}\right); \widehat{W}^{(i)} \right) \right)^\top \right\|_{2,\infty}$$

Using Lemma A.9 for each row:

$$\leq L_\sigma B_C^{(i)} \left\| f \left( g_{\text{tf-head}}^{(i)}\left(X; W^{1:i}\right); W^{(i)} \right)^\top - f \left( g_{\text{tf-head}}^{(i)}\left(X; \widehat{W}^{1:i}\right); \widehat{W}^{(i)} \right)^\top \right\|_{2,\infty}$$

By triangle inequality:

$$\leq L_\sigma B_C^{(i)} \left\| f \left( g_{\text{tf-head}}^{(i)}\left(X; W^{1:i}\right); W^{(i)} \right)^\top - f \left( g_{\text{tf-head}}^{(i)}\left(X; \widehat{W}^{1:i}\right); W^{(i)} \right)^\top \right\|_{2,\infty}$$

$$+ L_\sigma B_C^{(i)} \left\| f \left( g_{\text{tf-head}}^{(i)}\left(X; \widehat{W}^{1:i}\right); W^{(i)} \right)^\top - f \left( g_{\text{tf-head}}^{(i)}\left(X; \widehat{W}^{1:i}\right); \widehat{W}^{(i)} \right)^\top \right\|_{2,\infty}$$

By Lemma A.13 and A.14 and norm bounds on the matrices:

$$\leq L_\sigma B_C^{(i)} B_V^{(i)} \left(1 + 4 B_{QK}^{(i)}\right) \left\| g_{\text{tf-head}}^{(i)} \left(X; W^{1:i}\right)^\top - g\left(X; \widehat{W}^{1:i}\right)^\top \right\|_{2,\infty}$$

$$+ 2 L_\sigma B_C^{(i)} B_V^{(i)} \left\| \left(W_Q^{(i)} W_K^{(i)\top} - \widehat{W}_Q^{(i)} \widehat{W}_K^{(i)\top}\right) g_{\text{tf-head}}^{(i)} \left(X; \widehat{W}^{1:i}\right)^\top \right\|_{2,\infty}$$

$$+ L_\sigma B_C^{(i)} \left\| (W_V - \widehat{W}_V)^\top g_{\text{tf-head}}^{(i)} \left(X; \widehat{W}^{1:i}\right)^\top \right\|_{2,\infty}.$$

Combining the above gives us the desired result. $\qquad\square$

Lastly, we take account of the last linear weight and observe that,

**Lemma A.16.** *For any $W^{1:L+1}, \widehat{W}^{1:L+1}$ and $w, \widehat{w}$,*

$$\left| g_{\text{tf-scalar}} \left(X; W^{1:L+1}, w\right) - g_{\text{tf-scalar}} \left(X; \widehat{W}^{1:L+1}, \widehat{w}\right) \right|$$

$$\leq \|w\| \left\| g_{\text{tf-block}}^{(L+1)} \left(X; W^{1:L+1}\right)_{[\text{CLS}]} - g_{\text{tf-block}}^{(L+1)} \left(X; \widehat{W}^{1:L+1}\right)_{[\text{CLS}]} \right\| + \left| (w - \widehat{w})^\top g_{\text{tf-block}}^{(L+1)} \left(X; \widehat{W}^{1:L+1}\right)_{[\text{CLS}]} \right|.$$

*Proof.* Observe that,

$$\left| g_{\text{tf-scalar}} \left(X; W^{1:L+1}, w\right) - g_{\text{tf-scalar}} \left(X; \widehat{W}^{1:L+1}, \widehat{w}\right) \right|$$

$$= \left| w^\top g_{\text{tf-block}}^{(L+1)} \left(X; W^{1:L+1}\right)_{[\text{CLS}]} - \widehat{w}^\top g_{\text{tf-block}}^{(L+1)} \left(X; \widehat{W}^{1:L+1}\right)_{[\text{CLS}]} \right|$$

By triangle inequality:

$$\leq \left| w^\top \left( g_{\text{tf-block}}^{(L+1)} \left(X; W^{1:L+1}\right)_{[\text{CLS}]} - g_{\text{tf-block}}^{(L+1)} \left(X; \widehat{W}^{1:L+1}\right)_{[\text{CLS}]} \right) \right| + \left| (w - \widehat{w})^\top g_{\text{tf-block}}^{(L+1)} \left(X; \widehat{W}^{1:L+1}\right)_{[\text{CLS}]} \right|$$

Bounding the inner product by norms:

$$\leq \|w\| \left\| g_{\text{tf-block}}^{(L+1)} \left(X; W^{1:L+1}\right)_{[\text{CLS}]} - g_{\text{tf-block}}^{(L+1)} \left(X; \widehat{W}^{1:L+1}\right)_{[\text{CLS}]} \right\| + \left| (w - \widehat{w})^\top g_{\text{tf-block}}^{(L+1)} \left(X; \widehat{W}^{1:L+1}\right)_{[\text{CLS}]} \right|.$$

$$\square$$

### A.6.2 CONSTRUCTING THE COVER

The cover construction follows the standard recipe of composing covers per layer (as in Bartlett et al. (2017)).

**Theorem A.17.** *Let $\mathcal{F}_{\text{tf-scalar}}^{(L)}$ represent the class of functions of L-layer Transformer blocks satisfying the norm bounds (specified before) followed by linear layer on the* `[CLS]` *token. Then, for all $X^{(i)}$*

$$\log \mathcal{N}_\infty (\mathcal{F}_{\text{tf-scalar}}^{(L)}; \varepsilon; X^{(1)}, \dots, X^{(m)}, \|\cdot\|_2) \lesssim$$

$$\frac{\log(mT)}{\varepsilon^2} \times \left( B_w^{\frac{2}{3}} + \sum_{i=1}^{L} \alpha_i^{\frac{2}{3}} \left( d^{\frac{2}{3}} B_C^{\infty (i) \frac{2}{3}} + d^{\frac{2}{3}} \left(2 L_\sigma B_C^{(i)} B_V^{(i)} B_{QK}^{\infty (i)}\right)^{\frac{2}{3}} + k^{\frac{2}{3}} \left(L_\sigma B_C^{(i)} B_V^{\infty (i)}\right)^{\frac{2}{3}} \right) \right)^3$$

*where $\alpha_i = \prod_{j<i} L_\sigma B_C^{(j)} B_V^{(j)} (1 + 4 B_{QK}^{(j)})$.*

*Proof.* Our goal is to show that for every $\varepsilon > 0$, and collection of inputs $X^{(1)}, \dots, X^{(m)}$, there is a cover $\mathcal{C}$ of vectors in $\mathbb{R}^{(m)}$ such that for all $W^{1:L+1}$ and $w$ satisfying the norm bounds, there is some $v \in \mathcal{C}$ such that $\max_i |g_{\text{tf-scalar}}(X^{(i)}; W^{1:L+1}, w) - v| \leq \varepsilon$.

In each layer of the transformer, $W_Q^{(i)}$ and $W_K^{(i)}$ always appear together in the form $W_K^{(i)} W_Q^{(i)\top}$. Therefore, we will overload notation and define $W_{QK}^{(i)} : W_K^{(i)} W_Q^{(i)\top}$. Our cover $\mathcal{C}$ will be proper,

consisting of vectors of the form $(g_{\text{tf-scalar}}(X^{(i)}; \widehat{W}^{1:L+1}, \widehat{w}))_{i \in [m]}$. We will build the cover iteratively by finding finite collections of matrices $\widehat{\mathcal{W}}^{1:i}$ for each layer.

First observe that for any collection of $Z^{(1)}, \ldots, Z^{(m)} \in \mathbb{R}^{T \times d_1}$, and any $W, \widehat{W} \in \mathbb{R}^{d_1 \times d_2}$,

$$\max_{i \in [m]} \left\| W^\top Z^{(i)^\top} - \widehat{W}^\top Z^{(i)^\top} \right\|_{2,\infty} = \max_{i \in [m], t \in [T]} \left\| W^\top z_t^{(i)} - \widehat{W}^\top z_t^{(i)} \right\|.$$

This crucially allows us to aggregate over the samples and context length. In particular, we can apply Lemma 4.5 with the input vectors $(z_t^{(i)})_{i \in [m], t \in [T]}$; a total of $mT$ input vectors. Specifically, for any $\varepsilon$ and $\mathcal{W}(d_1, d_2, \alpha) := \{W \in \mathbb{R}^{d_1 \times d_2} \mid \|W^\top\|_{2,\infty} \leq \alpha\}$ with fixed $Z^{(i)}$ satisfying $\left\| Z^{(i)^\top} \right\|_{2,\infty} \leq 1$, Lemma 4.5 gives us such a cover.

First let us build a cover for one Transformer layer with inputs $Z^{(1)}, \ldots, Z^{(m)}$. We will begin with creating an $\varepsilon_V$-cover $\widehat{\mathcal{W}}_V$ for the function class of linear transformations given by $\mathcal{W}_V : \{W \in \mathbb{R}^{d \times k}, \|W^\top\|_{2,\infty} \leq \alpha, \|W\|_2 \leq s\}$ and $\varepsilon_{QK}$-cover $\widehat{\mathcal{W}}_{QK}$ for $\mathcal{W}_{QK} := \{W \in \mathbb{R}^{d \times d}, \|W\|_{2,\infty} \leq \beta, \|W\|_2 \leq r\}$ and inputs $Z^{(1)}, \ldots, Z^{(m)}$. For each pair of $\widehat{W}_V \in \widehat{\mathcal{W}}_V$ and $\widehat{W}_{QK} \in \widehat{\mathcal{W}}_{QK}$, we construct an $\varepsilon_C$-cover $\widehat{\mathcal{W}}_C(\widehat{W}_V, \widehat{W}_{QK})$ for $\mathcal{W}_C : \{W \in \mathbb{R}^{k \times d}, \|W\|_{2,\infty} \leq \gamma, \|W\|_2 \leq c\}$ and inputs $\left\{ \sigma \left( \Pi_{\text{norm}} \left( f \left( Z^{(i)}; \widehat{W}_V, \widehat{W}_{QK} \right) \right) \right) \right\}_{i=1}^m$. Our final cover is

$$\widehat{\mathcal{W}} := \left\{ (\widehat{W}_V, \widehat{W}_{QK}, \widehat{W}_C) : \widehat{W}_V \in \widehat{\mathcal{W}}_V, \widehat{W}_V \in \widehat{\mathcal{W}}_V, \widehat{W}_C \in \widehat{\mathcal{W}}_C(\widehat{W}_V, \widehat{W}_{QK}) \right\}.$$

Using Lemma A.15, we can show that $\widehat{\mathcal{W}}$ is an $\varepsilon$-cover for $g(\cdot; \{W_V, W_{QK}, W_C\})$ and inputs $Z^{(1)}, \ldots, Z^{(m)}$ where

$$\varepsilon = \varepsilon_C + 2L_\sigma cs\varepsilon_{QK} + L_\sigma c\varepsilon_V.$$

Using Lemma 4.5, the size of the cover is,

$$|\widehat{\mathcal{W}}| \leq |\widehat{\mathcal{W}}_V| |\widehat{\mathcal{W}}_{QK}| \max_{\substack{\widehat{W}_V \in \widehat{\mathcal{W}}_V \\ \widehat{W}_{QK} \in \widehat{\mathcal{W}}_{QK}}} \left| \widehat{\mathcal{W}}_C(\widehat{W}_V, \widehat{W}_{QK}) \right|$$

$$\implies \log |\widehat{\mathcal{W}}| \lesssim \left( \frac{k\alpha^2}{\varepsilon_V^2} + \frac{k\beta^2}{\varepsilon_{QK}^2} + \frac{d\gamma^2}{\varepsilon_C^2} \right) \log(mT).$$

We are now ready to inductively construct a cover for the deeper network. Suppose we have a $\varepsilon^{(i)}$-cover $\widehat{\mathcal{W}}^{1:i}$ for $g(\cdot; W^{1:i})$ on $X^{(1)}, \cdots, X^{(m)}$. We show how to construct an $\varepsilon^{(i+1)}$-cover for $g(\cdot; W^{1:i+1})$. For every element $\widehat{W}^{1:i} \in \widehat{\mathcal{W}}^{1:i}$ we construct a $\left( \varepsilon_C^{(i)} + 2L_\sigma B_C^{(i)} B_V^{(i)} \varepsilon_{QK}^{(i)} + L_\sigma B_C^{(i)} \varepsilon_V^{(i)} \right)$-cover $\widehat{\mathcal{W}}_i(\widehat{W}^{1:i})$ for the transformer layer (as above) on inputs $\left\{ g(X^{(j)}; \widehat{W}^{1:i}) \right\}_{j=1}^m$. Consider the cover

$$\widehat{\mathcal{W}}^{1:i+1} := \left\{ (\widehat{W}^{1:i}, \widehat{W}^{(i)}) : \widehat{W}^{1:i} \in \widehat{\mathcal{W}}^{1:i}, \widehat{W}^{(i)} \in \widehat{\mathcal{W}}_i(\widehat{W}^{1:i}) \right\}.$$

By Lemma A.15, this gives,

$$\varepsilon^{(i+1)} = L_\sigma B_C^{(i)} B_V^{(i)} (1 + 4B_{QK}^{(i)}) \varepsilon^{(i)} + \varepsilon_C^{(i)} + 2L_\sigma B_C^{(i)} B_V^{(i)} \varepsilon_{QK}^{(i)} + L_\sigma B_C^{(i)} \varepsilon_V^{(i)}.$$

The size of the cover is

$$|\widehat{\mathcal{W}}^{1:i+1}| \leq |\widehat{\mathcal{W}}^{1:i}| \max_{\widehat{W}^{1:i} \in \widehat{\mathcal{W}}^{1:i}} \left| \widehat{\mathcal{W}}_i(\widehat{W}^{1:i}) \right|.$$

Inductively applying this, we get

$$\varepsilon^{(L+1)} = \sum_{i=1}^L \left( \prod_{j<i} L_\sigma B_C^{(j)} B_V^{(j)} (1 + 4B_{QK}^{(j)}) \right) \left( \varepsilon_C^{(i)} + 2L_\sigma B_C^{(i)} B_V^{(i)} \varepsilon_{QK}^{(i)} + L_\sigma B_C^{(i)} \varepsilon_V^{(i)} \right)$$

$$= \sum_{i=1}^{L} \alpha_i \left( \varepsilon_C^{(i)} + 2L_\sigma B_C^{(i)} B_V^{(i)} \varepsilon_{QK}^{(i)} + L_\sigma B_C^{(i)} \varepsilon_V^{(i)} \right)$$

where $\alpha_i = \prod_{j<i} L_\sigma B_C^{(j)} B_V^{(j)} (1 + 4B_{QK}^{(j)})$.

The size of the cover is

$$\log \left( |\widehat{\mathcal{W}}^{1:L+1}| \right) \leq \sum_{i=1}^{L} \left( \frac{k^2 B_V^{\infty (i)^2}}{\varepsilon_V^{(i)^2}} + \frac{d^2 B_{QK}^{\infty (i)^2}}{\varepsilon_{QK}^{(i)^2}} + \frac{d^2 B_C^{\infty (i)^2}}{\varepsilon_C^{(i)^2}} \right) \log(mT).$$

Notice that the layer-norm maintains the norm bound on the inputs. Lastly, we need to cover the linear layer on the `[CLS]` token and compose it with the cover of $g^{1:L}$ (as before). Using Lemma A.10 and A.16, we can get the final $\varepsilon$-cover $\mathcal{C}$ with

$$\varepsilon = B_w \sum_{i=1}^{L} \alpha_i \left( \varepsilon_C^{(i)} + 2L_\sigma B_C^{(i)} B_V^{(i)} \varepsilon_{QK}^{(i)} + L_\sigma B_C^{(i)} \varepsilon_V^{(i)} \right) + \varepsilon_w$$

and size

$$\log |\mathcal{C}| \lesssim \frac{B_w^2 \log(m)}{\varepsilon_w^2} + \sum_{i=1}^{L} \left( \frac{k^2 B_V^{\infty (i)^2}}{\varepsilon_V^{(i)^2}} + \frac{d^2 B_{QK}^{\infty (i)^2}}{\varepsilon_{QK}^{(i)^2}} + \frac{d^2 B_C^{\infty (i)^2}}{\varepsilon_C^{(i)^2}} \right) \log(mT)$$

Using Lemma A.8, the size of the cover for fixed $\varepsilon$ gives us the desired result. $\qquad\square$

## B    Proofs for sparse function representation

### B.1    Setup

**Reductions from Boolean functions to Transformers.**    In order to establish our function approximation results, we must first define a canonical mapping between length-$T$ Boolean strings $b \in \{0,1\}^T$ and Transformer inputs $X \in \mathbb{R}^{T \times d}$. The key point (which has also been considered since the inception of the Transformer (Vaswani et al., 2017), and continues to be a crucial consideration in practice (Dosovitskiy et al., 2020)) is that the network's permutation-equivariant symmetry needs to be broken by assigning different embeddings to different indices of $b$. There are several possible natural choices here, which are all of practical interest:

- *Deterministic positional embeddings.* Fix positional embedding matrices $P \in \mathbb{R}^{T \times d}, E \in \mathbb{R}^{\{0,1\} \times d}$, and a special direction $v_{[\text{CLS}]} \in \mathbb{R}^d$, such that the $T + 3$ vectors $\{P_{t,:}\}_{t=1}^T \cup \{E_{j,:}\}_{j \in 0,1} \cup \{v_{[\text{CLS}]}\}$ are an approximately orthonormal basis for $\mathbb{R}^d$ (see below). The input to the Transformer is then $X = E_b + P$, where $E_b \in \mathbb{R}^{T \times d}$ such that $[E_b]_{t,:} = E_{b_t,:}$ for each $t \in [T]$. In the $f_{\text{tf-scalar}}$ formulation, we choose the auxiliary input $x_{[\text{CLS}]}$ to be the constant vector $v_{[\text{CLS}]}$. This closely matches applications of Transformers in NLP (Vaswani et al., 2017).

- *Trainable positional embeddings.* Like the above, but $P$ is a trainable parameter; we still require approximate orthogonality of $\{E_{j,:}\}_{j \in 0,1} \cup \{v_{[\text{CLS}]}\}$. It is also possible to consider the case where $E$ and $v_{[\text{CLS}]}$ are trainable (matching the way token embeddings are trained in practice). This becomes important in the regime of large vocabulary sizes that require embeddings to capture shared information between tokens; however, this is not necessary for our constructions, as we limit our consideration to binary tokens. This simplifies our constructions and improves statistical rates; additionally, it is a popular and well-studied alternative (Vaswani et al., 2017; Devlin et al., 2018; Radford et al., 2018; 2019; Brown et al., 2020).

- *Bag of vectors.* Fix a matrix $V \in \mathbb{R}^{T \times d}$ with approximately orthogonal rows (like the deterministic $P$), but choose the Transformer input

$$X := V \text{diag}(b).$$

This construction replaces positional embeddings with positional "indicator vectors" which can be swapped between any of the Transformer's input positions. It has the advantage of being symmetric with respect to permutation of the Transformer's input positions: it turns out that

$$f_{\text{tf-scalar}}(V \operatorname{diag}(b)) = f_{\text{tf-scalar}}(V \Pi \operatorname{diag}(b)),$$

for any $T \times T$ permutation matrix $\Pi$. It is also the most natural construction when considering the composition of sparse Boolean functions across multiple layers: a layer can output combinations of the basis rows $v_i$ for further function composition, like Boolean gates.

**Approximately orthonormal basis.** Each of the Boolean function approximation constructions will rely on a basis set of vectors, which will be used as positional embeddings (or the variable indices in the bag-of-vectors construction). We will fix a set of approximately orthonormal vectors $\{v_i : \|v_i\| = 1\}_{i=1}^{T'}$ in $\mathbb{R}^d$: for each $i \neq j$, we have $|v_i^\top v_j| \leq \Delta$. When $\Delta = 0$, the maximal $T'$ for which such a set exists is $d$; for $\Delta \in (0, \frac{1}{2})$, the Johnson-Lindenstrauss lemma (Johnson et al., 1986) implies that the maximal set of is of size $\exp(\Theta(d\Delta^2))$. For given choices of $d, \Delta$ and a maximal $\{v_1, \ldots, v_{T'}\}$, our construction is valid for contexts of length $T \leq T'$. For the special vectors $e_0, e_1, v_{[\text{CLS}]}$, we will assume that these are exactly orthogonal to the $v_i$ and each other, so that the $v_i$ must be a basis in dimension $d-1$ or $d-3$. This is for clarity only– it reduces the number of error terms to propagate through the analysis.

**Self-attention block.** In each construction (which specifies an input $X \in \mathbb{R}^{T \times d}$, we will specify the parameters $W_Q, W_K, W_V, W_C, w = e_1$ of a scalar-output Transformer $f_{\text{tf-scalar}}$, which takes an input $X \in \mathbb{R}^{(T+1) \times d}$; the auxiliary token input will be the constant vector $x_{[\text{CLS}]} := v_{[\text{CLS}]} \in \mathbb{R}^d$. The internal activation function $\sigma$ is chosen to be the identity. Summarizing, the functional form of $f_{\text{tf-scalar}} \in \mathcal{F}_{\text{tf-scalar}}$ in these constructions is

$$f_{\text{tf-scalar}}(X; W_Q, W_K, W_V, W_C, e_1) = \operatorname{softmax}\left(v_{[\text{CLS}]}^\top W_Q W_K^\top X^\top\right) X W_V W_C e_1.$$

In the intermediate lemmas, it will also be useful to consider the corresponding attention head output

$$f_{\text{tf-head}}(X; W_Q, W_K, W_V, W_C) = \operatorname{softmax}\left(v_{[\text{CLS}]}^\top W_Q W_K^\top X^\top\right) X W_V W_C,$$

and its projections $f_{\text{tf-head}} \circ \Pi_{d_{\text{proj}}}$ onto the first $d_{\text{proj}}$ coordinates.

**Feedforward networks.** We establish some notation for feedforward networks. An $L$-layer feedforward network, with activation function $\sigma : \mathbb{R} \to \mathbb{R}$ and dimensions $d_1, \ldots, d_{L+1}$, is parameterized by weight matrices $W_i \in \mathbb{R}^{d_{i+1} \times d_i}$, and maps $x \in \mathbb{R}^{d_1}$ to $y \in \mathbb{R}^{d_{L+1}}$, by the iterative equations

$$y_1^\top := \sigma(x^\top W_1),$$
$$y_{i+1}^\top := \sigma(y_i^\top W_i), \qquad i = 1, \ldots, L-1,$$
$$f_{\text{mlp}}(x; W_1, \ldots, W_L)^\top = y^\top := y_L^\top W_L.$$

When $d_{L+1} = 1$, we will use the notation $w$ instead of $W_L$. It will be convenient to incorporate *bias* weights by introducing an extra input coordinate $W_i \in \mathbb{R}^{(d_{i+1}+1) \times d_i}$, and augmenting the linear function accordingly:

$$y_i^\top W_i \mapsto [y_i^\top \ 1] W_i.$$

**Self-attention composed with a feedforward network.** The full definition of the Transformer layer composes a self-attention layer ($f_{\text{tf-layer}} : \mathbb{R}^{T \times d} \to \mathbb{R}^{T \times d}$) with a position-wise feedforward network ($f_{\text{mlp}} : \mathbb{R}^d \to \mathbb{R}^d$). We will use this combination of modules to establish our function approximation results: $f_{\text{tf-layer}}$ acts as a sparse bottleneck, while $f_{\text{mlp}}$ approximates an arbitrary function of the selected coordinates. For our single-layer constructions, it is most convenient to establish notation for a scalar-output Transformer with a feedforward network. To this end, define $\mathcal{F}_{\text{tf+mlp}}$ to be the function class with the same Score, Norm, $\phi_{\text{in}}$ functions as in $\mathcal{F}_{\text{tf-scalar}}$ (thus, the same parameters $W_Q, W_K, W_V$), with identity activation function, but a feedforward neural network replacing the linear $\phi_{\text{out}}$ and $w$. Concretely, with $L = 3$ and the ReLU activation function $(\cdot)_+$, $\mathcal{F}_{\text{tf+mlp}}$ contains functions of the form

$$f_{\text{tf+mlp}}(X; \theta) = \left((y^\top W_1)_+ W_2\right)_+ w,$$

$$y = \operatorname{softmax}\left(v_{[\text{CLS}]}^\top W_Q W_K^\top X^\top\right) X W_V W_C w,$$

with parameters $\theta := (W_Q, W_K, W_V, W_C, W_1, W_2, w)$.

**Multiple self-attention heads.** The final component we will need for the function approximation setup is multi-headed self-attention. We will extend the definition of the single-headed $f_{\text{tf-head}}$ to

$$f_{\text{tf-heads}}\left(X; \left\{W_Q^{[h]}, W_K^{[h]}, W_V^{[h]}, W_C^{[h]}\right\}_{h=1}^H\right) := \sum_{h=1}^H f_{\text{tf-head}}\left(X; W_Q^{[h]}, W_K^{[h]}, W_V^{[h]}, W_C^{[h]}\right),$$

and substitute this definition into $f_{\text{tf+mlp}}$ when discussing a multi-headed construction.

**Classes and properties of Boolean functions.** We will call a Boolean function $f : \{0,1\}^T \to \mathcal{Y}$ $\mathcal{I}$-*sparse* if it only depends on a fixed subset $\mathcal{I} \subseteq [T]$ of its inputs:

$$b_i = b_i' \quad \forall i \in \mathcal{I} \implies f(b) = f(b').$$

Overloading notation, if $\mathcal{I} = s$, we will also call $f$ $s$-sparse. We will call an $\mathcal{I}$-sparse Boolean function $f$ *symmetric* if its value is invariant under permutation of the indices in $\mathcal{I}$:

$$|\{i \in \mathcal{I} : b_i = 1\}| = |\{i \in \mathcal{I} : b_i' = 1\}| \implies f(b) = f(b').$$

Further, we will call an $\mathcal{I}$-sparse real-valued symmetric Boolean function $f : \{0,1\}^T \to \mathcal{Y}$ *monotone* if $f(b)$ is monotonically increasing in $r := |\{i \in \mathcal{I} : b_i = 1\}|$. If, for some $\gamma > 0$, it holds that $f(r+1) \geq f(r) + \gamma$ for each $r = 0, \ldots, s-1$, we call $f$ $\gamma$-*strictly monotone*. A vector-valued $\mathcal{I}$-sparse Boolean function $f : \{0,1\}^T \to \mathbb{R}^{d_f}$ is $\gamma$-*injective* if

$$\|f(b) - f(b')\|_\infty \geq \gamma$$

for each $b, b'$ that differ at some position $i \in \mathcal{I}$; $f$ is called $B$-*bounded* if $\|f(b)\|_\infty \leq B$ for all $b \in \{0,1\}^T$.

**Uniform approximation.** For some $\varepsilon \geq 0$ and a function $f : \{0,1\}^T \to \mathbb{R}^d$, we say that $\widehat{f} \in \mathcal{F}$ $\varepsilon$-*uniformly approximates* $f$ under the mapping $b \mapsto X(b)$ if

$$\left\|\widehat{f}(X(b)) - f(b)\right\|_\infty \leq \varepsilon, \qquad \forall b \in \{0,1\}^T.$$

## B.2 Results

We give an overview of the function approximation results under each input mapping $b \mapsto X(b)$, as a multi-part proposition:

**Proposition B.1** (Sparse variable creation with Transformers). *The function classes* $\mathcal{F}_{\text{tf-scalar}}, \mathcal{F}_{\text{tf+mlp}}$ *contain the following classes of sparse Boolean functions:*

- *Deterministic positional embeddings:* For any $\mathcal{I}$, $\mathcal{F}_{\text{tf-scalar}}$ can approximate a particular monotone symmetric $\mathcal{I}$-sparse $f$, with Transformer weight norm bounds from the real-valued construction in Lemma B.2. $\mathcal{F}_{\text{tf+mlp}}$ with 1 head can exactly represent any symmetric $s$-sparse $f$, with the same bounds on Transformer weight norms, and feedforward network weight norms scaling as $O(\text{poly}(s))$. $\mathcal{F}_{\text{tf+mlp}}$ with $s$ heads can exactly represent any $s$-sparse $f$, with Transformer weight norm bounds from the vector-valued construction in Lemma B.2, and feedforward network weight norms scaling as $O(\text{poly}(s) \cdot 2^s)$.

- *Trainable positional embeddings:* For any $\mathcal{I}$, $\mathcal{F}_{\text{tf-scalar}}$ can approximate a particular monotone symmetric $\mathcal{I}$-sparse $f$, with positional embedding and Transformer weight norm bounds from the real-valued construction in Lemma B.3. $\mathcal{F}_{\text{tf+mlp}}$ with 1 head can exactly represent any symmetric $s$-sparse $f$, with the same bounds on $P$ and Transformer weight norms, and feedforward network weight norms scaling as $O(\text{poly}(s))$. $\mathcal{F}_{\text{tf+mlp}}$ with $s$ heads can exactly represent any sparse $f$, with $P$ and Transformer weight norm bounds from the vector-valued construction in Lemma B.3, and feedforward network weight norms scaling as $O(\text{poly}(s) \cdot 2^s)$.

- *Bag of vectors:* For any $\mathcal{I}$, $\mathcal{F}_{\text{tf-scalar}}$ can approximate a particular monotone symmetric $\mathcal{I}$-sparse $f$, with Transformer weight norms from Lemma B.4. $\mathcal{F}_{\text{tf+mlp}}$ with 1 head can represent any symmetric $s$-sparse $f$, with the same Transformer weight norm bounds, and feedforward network weight norms scaling as $O(\text{poly}(s))$. $\mathcal{F}_{\text{tf+mlp}}$ with 1 head can also exactly represent any $s$-sparse $f$, with the same bounds on Transformer weight norms, and feedforward network weight norms scaling as $O(\text{poly}(s) \cdot 2^s)$.

The formal statements are obtained by $(\gamma/4)$-uniformly approximating a $\gamma$-strictly monotone or $\gamma$-injective function with self-attention alone (Lemmas B.2, B.3, B.4), then applying a robust universal function representation construction (Lemmas B.5, B.6) appropriately. They are organized as follows:

**Lemma B.2** (Deterministic $P$, no MLP). *Suppose $X(b) = P + E_b$ with deterministic $P$. Let $\mathcal{I} \subseteq [T]$ such that $|\mathcal{I}| = s \leq d, k$, and $\Delta < 1/s$. Then, for all $0 < \gamma \leq 1$, there exists a 1-bounded, $(2/s)$-strictly monotone symmetric $\mathcal{I}$-sparse Boolean function $g_{\mathcal{I}} : \{0,1\}^T \to \mathbb{R}$ and Transformer head parameters such that $f_{\text{tf-scalar}}(X(b); W_Q, W_K, W_V, W_C, w = e_1)$ $(\gamma/4)$-uniformly approximates $g_{\mathcal{I}}$. The norms satisfy*

$$\|W_Q\|_F \leq \frac{\log\left(\frac{8T}{\gamma}\right)}{1 - s\Delta}, \qquad \|W_K\|_F \leq s, \qquad \|W_V\|_F \leq 2, \qquad \|W_C\|_F \leq 1.$$

*Also, there exists a 1-bounded, 2-injective $\mathcal{I}$-sparse Boolean function $g'_{\mathcal{I}} : \{0,1\}^T \to \mathbb{R}^s$ and $s$-headed Transformer parameters such that $f_{\text{tf-head}}\left(X(b); \left\{W_Q^{[h]}, W_K^{[h]}, W_V^{[h]}, W_C^{[h]}\right\}_{h=1}^s\right) \circ \Pi_s$ uniformly approximates $g'_{\mathcal{I}}$. The norms of each head satisfy*

$$\left\|W_Q^{[h]}\right\|_F \leq \frac{\log\left(\frac{8T}{\gamma}\right)}{1 - s\Delta}, \qquad \left\|W_K^{[h]}\right\|_F \leq 1, \qquad \left\|W_V^{[h]}\right\|_F \leq 2, \qquad \left\|W_C^{[h]}\right\|_F \leq 1.$$

**Lemma B.3** (Trainable $P$, no MLP). *Suppose $X(b) = P + E_b$ with trainable $P$. Let $\mathcal{I} \subseteq [T]$ such that $|\mathcal{I}| = s \leq d, k$. Then, for any $0 < \gamma \leq 1$, and with the same $g_{\mathcal{I}}$ as in Lemma B.2, there exists $P$ and Transformer head parameters such that $f_{\text{tf-scalar}}(X(b); W_Q, W_K, W_V, W_C, w = e_1)$ $(\gamma/4)$-uniformly approximates $g_{\mathcal{I}}$. The norms satisfy*

$$\left\|P^\top\right\|_{2,1} \leq s, \qquad \|W_Q\|_F \leq \log\left(\frac{8T}{\gamma}\right), \qquad \|W_K\|_F \leq 1, \qquad \|W_V\|_F \leq 2, \qquad \|W_C\|_F \leq 1.$$

*Also, for the same $g'_{\mathcal{I}}$ as in Lemma B.2, there exists $P$ and $s$-headed Transformer parameters such that $f_{\text{tf-head}}\left(X(b); \left\{W_Q^{[h]}, W_K^{[h]}, W_V^{[h]}, W_C^{[h]}\right\}_{h=1}^s\right) \circ \Pi_s$ uniformly approximates $g'_{\mathcal{I}}$. The norms of each head satisfy*

$$\left\|P^\top\right\|_{2,1} \leq s, \qquad \left\|W_Q^{[h]}\right\|_F \leq \log\left(\frac{8T}{\gamma}\right), \qquad \left\|W_K^{[h]}\right\|_F \leq 1, \qquad \left\|W_V^{[h]}\right\|_F \leq 2, \qquad \left\|W_C^{[h]}\right\|_F \leq 1.$$

**Lemma B.4** (Bag of vectors, no MLP). *Suppose $X(b) = V + \text{diag}(b)$. Let $\mathcal{I} \subseteq [T]$ such that $|\mathcal{I}| = s \leq d, k$, and $\Delta < 1/s$. Then, for all $s\Delta < \gamma < 1$, there exists an $s$-bounded, $(1/s)$-strictly monotone symmetric $\mathcal{I}$-sparse Boolean function $g_{\mathcal{I}} : \{0,1\}^T \to \mathbb{R}$ and Transformer head parameters such that $f_{\text{tf-scalar}}(X(b); W_Q, W_K, W_V, W_C, w = e_1)$ $(\gamma/4)$-uniformly approximates $g_{\mathcal{I}}$. The norms satisfy*

$$\|W_Q\|_F \leq \frac{\log\left(\frac{8Ts(1+\Delta)}{\gamma - s\Delta}\right)}{1 - s\Delta}, \qquad \|W_K\|_F \leq s + 1, \qquad \|W_V\|_F \leq 2s, \qquad \|W_C\|_F \leq s.$$

*Also, there exists a 1-bounded, $(1/s)$-injective $\mathcal{I}$-sparse Boolean function $g'_{\mathcal{I}} : \{0,1\}^T \to \mathbb{R}^s$ and Transformer head parameters such that $f_{\text{tf-head}}(X(b); W_Q, W_K, W_V, W_C) \circ \Pi_s$ uniformly approximates $g'_{\mathcal{I}}$. The norms satisfy the same bounds as above.*

**Lemma B.5** (Monotone to symmetric functions via MLP). *Let $f : \{0,1\}^T \to \mathbb{R}$ be any real-valued symmetric $s$-sparse Boolean function with index set $\mathcal{I}$. Let $W_Q, W_K, W_V, W_C$ be the parameters of a function*

$$f_{\text{tf-head}}(X; W_Q, W_K, W_V, W_C) := \text{softmax}\left(v_{[\text{CLS}]}^\top W_Q W_K^\top X^\top\right) X W_V W_C,$$

*and let $\Pi_1 : \mathbb{R}^d \to \mathbb{R}$ be the projection onto the first coordinate. Suppose that under some mapping $b \mapsto X(b)$, $f_{\text{tf-head}} \circ \Pi_s$ $(\gamma/4)$-uniformly approximates a $B$-bounded $\gamma$-strictly monotone symmetric $\mathcal{I}$-sparse Boolean function $g : \{0,1\}^T \to \mathbb{R}$, for some $\gamma$. Then, there exists a function $f_{\text{tf+mlp}} \in \mathcal{F}_{\text{tf+mlp}}$ with the same weights $W_Q, W_K, W_V, W_C$, and 3-layer feedforward network weights $W_1, W_2, w$, such that*

$$f_{\text{tf+mlp}}(X(b)) = f(b), \qquad \forall b \in \{0,1\}^T,$$

*with dimensions* $(d_2, d_3) = (4(s+1), 2(s+1))$ *and weight norms satisfying*

$$\|W_1\|_\infty \le \frac{8 \max(1, B)}{\gamma}, \qquad \|W_2\|_\infty \le \frac{8s}{\gamma}, \qquad \|w\|_\infty \le \max_{b \in \{0,1\}^T} |f(b)|.$$

**Lemma B.6** (Injective to arbitrary functions via MLP)**.** *Let* $f : \{0,1\}^T \to \mathbb{R}$ *be any real-valued* $s$-*sparse Boolean function with index set* $\mathcal{I}$ *such that* $|\mathcal{I}| = s \le d$. *Let* $W_Q, W_K, W_V, W_C$ *be the parameters of a function*

$$f_{\text{tf-head}}(X; W_Q, W_K, W_V, W_C) := \text{softmax}\left(v_{[\text{CLS}]}^\top W_Q W_K^\top X^\top\right) X W_V W_C,$$

*and let* $\Pi_s : \mathbb{R}^d \to \mathbb{R}^s$ *be the projection onto the first* $s$ *coordinates. Suppose that under some mapping* $b \mapsto X(b)$, $f_{\text{tf-head}} \circ \Pi_s$ $(\gamma/4)$-*uniformly approximates a* $\gamma$-*injective function* $g : \{0,1\}^T \to \mathbb{R}^s$ *satisfying* $\|g(b)\|_\infty \le B$. *Then, there exists a function* $f_{\text{tf+mlp}} \in \mathcal{F}_{\text{tf+mlp}}$ *with the same weights* $W_Q, W_K, W_V, W_C$, *and 3-layer feedforward network weights* $W_1, W_2, w$, *such that*

$$f_{\text{tf+mlp}}(X(b)) = f(b), \qquad \forall b \in \{0,1\}^T,$$

*with dimensions* $(d_2, d_3) = (4s2^s, 2 \cdot 2^s)$ *and weight norms satisfying*

$$\|W_1\|_\infty \le \frac{8 \max(1, B)}{\gamma}, \qquad \|W_2\|_\infty \le \frac{8s}{\gamma}, \qquad \|w\|_\infty \le \max_{b \in \{0,1\}^T} |f(b)|.$$

### B.3 USEFUL LEMMAS

We will use a construction which approximates a "hard selection" of $s$ indices using the softmax mixture; for this, we will need to quantify the approximation error when the inputs to the softmax function are bounded.

**Lemma B.7** (Softmax truncation)**.** *Let* $z \in (\mathbb{R} \cup \{-\infty\})^T$ *such that* $z_t \ge R$ *for each* $1 \le t \le s$, *and* $z_t \le 0$ *for each* $s+1 \le t \le T$. *Define* $z' \in (\mathbb{R} \cup \{-\infty\})^T$ *so that* $z'_t = z_t$ *for* $1 \le t \le s$, *and* $z_t = -\infty$ *for* $s+1 \le t \le T$. *Then, letting* $e^{-\infty} = 0$ *in the definition of* $\text{softmax}(\cdot)$, *we have*

$$\|\text{softmax}(z') - \text{softmax}(z)\|_1 \le 2\frac{T-s}{s\exp(R)} < \frac{2T}{\exp(R)}.$$

*Proof.* We have

$$\|\text{softmax}(z') - \text{softmax}(z)\|_1 = \sum_{t=1}^s \exp(z_t) \left(\frac{1}{\mathbf{1}^\top \exp(z')} - \frac{1}{\mathbf{1}^\top \exp(z)}\right) + \sum_{t=s+1}^T \frac{\exp(z_t)}{\mathbf{1}^\top \exp(z)}.$$

The first summation is equal to

$$1 - \frac{\mathbf{1}^\top \exp(z')}{\mathbf{1}^\top \exp(z)} \le \frac{T-s}{s\exp(R)},$$

while the same upper bound holds for the second summation, since each term is at most $\frac{1}{s\exp(R)}$. $\square$

Our results on approximating arbitrary sparse Boolean functions will depend on a generic construction for robustly approximating an arbitrary function $f : \mathbb{R}^d \to \mathbb{R}$ with a feedforward neural network. For simplicity of presentation, we use a standard[6] 3-layer ReLU network construction, which *exactly* represents a piecewise constant function in specified regions.

**Lemma B.8** (Exact function representation with a 3-layer ReLU net)**.** *Let* $f : \mathbb{R}^{d_f} \to \mathbb{R}$, *and let* $x_1, \ldots, x_n \in \mathbb{R}^{d_f}$ *such that* $\|x_i\|_\infty \le B$ *for each* $i \in [n]$, $\|x_i - x_j\|_\infty \ge 4\delta$ *for each* $i \ne j \in [n]$. *Then, there is a 3-layer feedforward network with ReLU activations, with parameters* $W_1 \in \mathbb{R}^{(d_f+1) \times d_2}, W_2 \in \mathbb{R}^{(d_2+1) \times d_3}, w \in \mathbb{R}^{d_3}$[7], *such that*

$$f_{\text{mlp}}(x_i + z) = f(x_i)$$

*for all* $i \in [n]$ *and* $\|z\|_\infty \le \delta$, *where* $\text{ReLU}(x) := x_+ = \max(0, x)$ *is applied entrywise, with*

$$d_2 = 4nd_f, \quad d_3 = 2n,$$

$$\|W_1\|_\infty \le \frac{\max(1, B)}{\delta}, \quad \|W_2\|_\infty \le \frac{d_f}{\delta}, \quad \|w\|_\infty \le \max_{i \in [n]} |f(x_i)|.$$

---

[6]For example, this follows from the discussion in Chapter 4 of (Nielsen, 2015).

[7]Here, $W_1, W_2$ have bias terms; $w$ does not.

*Proof.* First, we construct a one-dimensional "bump" function basis, and propagate the Lipschitz constants. A threshold function with a linear "ramp" of width $\delta$ can be obtained from a linear combination of 2 ReLU functions:

$$\nu_\delta(x) := (x/\delta + 1)_+ - (x/\delta)_+ = \begin{cases} 0 & x \leq -\delta \\ x/\delta + 1 & -\delta \leq x \leq 0 \\ 1 & x \geq 0 \end{cases}.$$

Next, we construct the bump function

$$\psi_\delta(x) := \nu_\delta(x) - \nu_\delta(2\delta - x).$$

By this construction, we have $\psi_\delta(x) = 1$ for $0 \leq x \leq 2\delta$ and $\psi_\delta(x) = 0$ for $x \leq -\delta$ and $x \geq 3\delta$, interpolating linearly on $[-\delta, 0]$ and $[2\delta, 3\delta]$. Next, define

$$\psi_\delta(x; x_0) := \psi_\delta(x - x_0 + \delta)$$

$$= \left( \frac{x - x_0}{\delta} + 2 \right)_+ - \left( \frac{x - x_0}{\delta} + 1 \right)_+ - \left( \frac{x_0 - x}{\delta} + 2 \right)_+ + \left( \frac{x_0 - x}{\delta} + 1 \right)_+$$

so that $\psi_\delta(x; x_0) = 1$ for $|x - x_0| \leq \delta$, $\psi_\delta(x; x_0) = 0$ for $|x - x_0| \geq 2\delta$.

We construct the first layer $W_1 \in \mathbb{R}^{(d+1) \times (4nd)}$ using these bump functions: indexing the $4nd$ dimension by $(h \in [4], i \in [n], j \in [d])$, we construct

$$[W_1]_{:,(1,i,:)} := \begin{bmatrix} \frac{1}{\delta} I \\ -\frac{x_i}{\delta} + 2 \cdot \mathbf{1}^\top \end{bmatrix}, \qquad [W_1]_{:,(2,i,:)} := \begin{bmatrix} \frac{1}{\delta} I \\ -\frac{x_i}{\delta} + \mathbf{1}^\top \end{bmatrix},$$

$$[W_1]_{:,(3,i,:)} := \begin{bmatrix} -\frac{1}{\delta} I \\ \frac{x_i}{\delta} + 2 \cdot \mathbf{1}^\top \end{bmatrix}, \qquad [W_1]_{:,(4,i,:)} := \begin{bmatrix} -\frac{1}{\delta} I \\ \frac{x_i}{\delta} + \mathbf{1}^\top \end{bmatrix},$$

so that

$$\left( [x \; 1]^\top \left[ [W_1]_{j,(1,i,:)} \; [W_1]_{j,(2,i,:)} \; [W_1]_{j,(3,i,:)} \; [W_1]_{j,(4,i,:)} \right] \right)_+ [1 \; -1 \; -1 \; 1]^\top = \psi_\delta(x; [x_i]_j).$$

The second layer is used to construct $n$ activations which are indicators of whether $x$ is in the neighborhood of each $x_i$. For each $x_i$, we will simply average the $d_f$ one-dimensional indicators for each coordinate, and implement a threshold function $\nu_{\delta/d_f}(1 - x)$. We choose $W_2 \in \mathbb{R}^{(4nd_f + 1) \times (2n)}$, with the $4nd_f + 1$ dimension indexed by $(h, i, j)$ and an extra bias dimension $\perp$, and the $2n$ dimension indexed by $(h' \in \{1, 2\}, i' \in [n])$ so that

$$[W_2]_{(h,i,:),(h',i')} := [1 \; -1 \; -1 \; 1]_h \cdot \mathbb{1}[i = i'] \cdot \frac{1}{\delta} \cdot \mathbf{1},$$

$$[W_2]_{\perp,(1,i')} := 1 - \frac{d_f}{\delta}, \qquad [W_2]_{\perp,(2,i')} := -\frac{d_f}{\delta}.$$

Finally, the third (output) layer $w \in \mathbb{R}^{2n}$, with dimensions indexed by $(h \in \{1, 2\}, i \in [n])$, multiplies the indicators of each $x_i$ by the desired $f(x_i)$:

$$w_{(1,i)} := f(x_i), \quad w_{(2,i)} := -f(x_i).$$

For any $x_0 \in \mathbb{R}^{d_f}$, let $B_\delta(x_0)$ be the set of $x$ such that $\|x - x_0\|_\infty \leq \delta$. By this construction, for each $x \in B_\delta(x_i)$, we have $f(x) = x_i$, as required. $\qquad \square$

Note that we use 3-layer ReLU networks for function approximation in order to minimize the introduction of unnecessary notation. Some remarks:

- It would be routine to replace this construction with any architecture which can represent an arbitrary function *approximately* (Hornik et al., 1989; Cybenko, 1989); this includes the 2-layer feedforward networks (and nonlinear activations other than the ReLU) which are typically used by Transformers in practice.

- It is possible to embed this construction in $f_{\text{tf+mlp}}$ with a 2-layer ReLU network, by using $(W_C, W_1, W_2)$ and introducing a nonlinearity after $W_C$, without changing the results.

- When $d_f = 1$, $W_2$ is unnecessary (one can represent $f$ directly using the bump function basis).

### B.4 PROOFS

Throughout the constructions in each case, we will refer to standard coordinate bases in several spaces:

- $E_0, E_1 \in \mathbb{R}^d$ denote the embeddings of the $0, 1$ tokens $E_{0,:}, E_{1,:}$.

- $e_1^{(k)}, \dots, e_k^{(k)}$ denotes the standard basis in $\mathbb{R}^k$.

- $e_i^{(d)}$ denotes the standard basis in $\mathbb{R}^d$.

- $e_1^{(T)}, \dots, e_T^{(T)}, e_{[\text{CLS}]}^{(T)}$ denotes the standard basis in $\mathbb{R}^{T+1}$ with the special $[\text{CLS}]$ index.

- Recall that the $v_i$ form a $\Delta$-approximate orthonormal basis for $\mathbb{R}^d$, $v_{[\text{CLS}]}, e_0, e_1$ are exactly orthogonal to each of them as well as each other, and $d$ is chosen such that these conditions can be met.

Let $n(i)$ be a unique bijection between $\mathcal{I}$ and $[s]$. Let $v_{\mathcal{I}} := \sum_{i \in \mathcal{I}} v_i$.

**Approximate vector equality.** We will use $u \approx_\varepsilon v$ to denote that two vectors $u, v \in \mathbb{R}^{d_u}$ satisfy $\|u - v\|_\infty \le \varepsilon$.

*Proof of Lemma B.2.* We construct attention heads such that the softmax mixture always selects the indices in $\mathcal{I}$.

**Single head, deterministic $P$.** We seek to approximate the 1-bounded, $(2/s)$-strictly monotone function

$$\frac{1}{s} \sum_{i \in \mathcal{I}} \chi_i,$$

where $\chi_i = +1$ if $b_i = 0$ and $-1$ if $b_i = 1$. Set

$$W_Q := R v_{[\text{CLS}]} e_1^{(k)\top}, \quad W_K := v_{\mathcal{I}} e_1^{(k)\top}, \quad W_V := (E_0 - E_1) e_1^{(k)\top}, \quad W_C := e_1^{(k)} e_1^{(d)\top},$$

where $R$ will be chosen later. Then, by approximate orthogonality,

$$v_{[\text{CLS}]}^\top W_Q W_K^\top X^\top = v_{[\text{CLS}]}^\top W_Q W_K^\top (P + E_b)^\top = v_{[\text{CLS}]}^\top W_Q W_K^\top P^\top \approx_{Rs\Delta} R \sum_{i \in \mathcal{I}} e_i^{(T)\top}.$$

By Lemma B.7,

$$\left\| \text{softmax}\left( v_{[\text{CLS}]}^\top W_Q W_K^\top X^\top \right) - \frac{1}{s} \sum_{i \in \mathcal{I}} e_i^{(T)\top} \right\|_1 \le \frac{2T}{\exp(R - 2Rs\Delta)}.$$

Finally, we have

$$X W_V W_C = E_b W_V W_C = \left( \sum_{i \in [T]} \chi_i e_i^{(T)} \right) e_1^{(d)\top},$$

so that by Hölder's inequality,

$$f_{\text{tf-head}}(X) \circ \Pi_1 = \text{softmax}\left( v_{[\text{CLS}]}^\top W_Q W_K^\top X^\top \right) X W_V W_C e_1^{(d)} \approx_{\frac{2T}{\exp(R-2Rs\Delta)}} \frac{1}{s} \sum_{i \in \mathcal{I}} \chi_i.$$

To get $(\gamma/4)$-uniform approximation, we choose

$$R = \frac{\log\left( \frac{8T}{\gamma} \right)}{1 - s\Delta}.$$

**Multiple heads, deterministic $P$.** For $h = 1, \dots, s$, and the same $R$ as above:

$$W_Q^{[h]} := R v_{[\text{CLS}]} e_1^{(k)\top}, \quad W_K^{[h]} := v_{n^{-1}(h)} e_1^{(k)\top}, \quad W_V^{[h]} := (E_0 - E_1) e_2^{(k)\top}, \quad W_C^{[h]} := e_1^{(k)} e_h^{(d)\top}.$$

This is the same construction as above, but each head only selects one of the coordinates in $\mathcal{I}$. Thus, by the same analysis,

$$f_{\text{tf-head}}(X) \circ \Pi_s \approx_{\frac{2T}{\exp(R - 2Rs\Delta)}} \sum_{i \in \mathcal{I}} \chi_i e_{n(i)}^{(d)}.$$

This function is clearly 1-bounded and 2-injective. $\qquad\square$

*Proof of Lemma B.3.* The constructions closely follow Lemma B.2, but are simpler.

**Single head, trainable** $P$**.** For each $i \in \mathcal{I}$, set the trainable positional embeddings to be

$$P_{i,:} := \begin{cases} v_1 & i \in \mathcal{I} \\ 0 & \text{otherwise} \end{cases}.$$

Set

$$W_Q := R v_{[\text{CLS}]} e_1^{(k)\top}, \quad W_K := v_1 e_1^{(k)\top}, \quad W_V := (E_0 - E_1) e_1^{(k)\top}, \quad W_C := e_1^{(k)} e_1^{(d)\top}.$$

Now, we have (with equality)

$$v_{[\text{CLS}]}^\top W_Q W_K^\top X^\top = R \sum_{i \in \mathcal{I}} e_i^{(T)\top},$$

so that Lemma B.7 gives

$$\left\| \text{softmax}\left( v_{[\text{CLS}]}^\top W_Q W_K^\top X^\top \right) - \frac{1}{s} \sum_{i \in \mathcal{I}} e_i^{(T)\top} \right\|_1 \le \frac{2T}{\exp(R)}.$$

Like before, we have

$$f_{\text{tf-head}}(X) \circ \Pi_1 = \text{softmax}\left( v_{[\text{CLS}]}^\top W_Q W_K^\top X^\top \right) X W_V W_C e_1^{(d)} \approx_{\frac{2T}{\exp(R)}} \frac{1}{s} \sum_{i \in \mathcal{I}} \chi_i.$$

To get $(\gamma/4)$-uniform approximation, we choose

$$R = \log\left( \frac{8T}{\gamma} \right).$$

**Multiple heads, trainable** $P$**.** For each $i \in \mathcal{I}$, set the trainable positional embeddings to be

$$P_{i,:} := \begin{cases} e_{n(i)}^{(d)} & i \in \mathcal{I} \\ 0 & \text{otherwise} \end{cases}.$$

For $h = 1, \ldots, s$, and the same $R$ as above:

$$W_Q^{[h]} := R v_{[\text{CLS}]} e_1^{(k)\top}, \quad W_K^{[h]} := e_h^{(d)} e_1^{(k)\top}, \quad W_V^{[h]} := (E_0 - E_1) e_1^{(k)\top}, \quad W_C^{[h]} := e_1^{(k)} e_h^{(d)\top}.$$

This is the same construction as above, but each head only selects one of the coordinates in $\mathcal{I}$. Thus, by the same analysis,

$$f_{\text{tf-head}}(X) \circ \Pi_s \approx_{\frac{2T}{\exp(R)}} \sum_{i \in \mathcal{I}} \chi_i e_{n(i)}^{(d)}.$$

$\qquad\square$

*Proof of Lemma B.4.* This input mapping does not use position embeddings, and does not need multiple heads to implement arbitrary (non-symmetric) functions. The constructed monotone and injective functions are slightly different, but the proof strategy is very similar. The key difference is that the softmax mixture is uniform only on the positions $i \in \mathcal{I}$ where $b_i = 1$.

**Bag of vectors, scalar output.** The function we will approximate is defined as follows:

$$g_{\mathcal{I}}(r) := \frac{r - s}{r + 1}, \quad \text{where } r = \sum_{i \in \mathcal{I}} b_i, \quad s = |\mathcal{I}|.$$

Note that this function is $(1/s)$-strictly monotone, and has absolute value bounded by $s$. Set

$$W_Q := Rv_{[\text{CLS}]}e_1^{(k)\top}, \quad W_K := (v_\mathcal{I} + v_{[\text{CLS}]})e_1^{(k)\top},$$

$$W_V := \sum_{i\in\mathcal{I}} v_i e_{n(i)}^{(k)\top} - v_{[\text{CLS}]}\left(\sum_{i\in\mathcal{I}} e_{n(i)}^{(k)}\right)^\top, \quad W_C := \sum_{i\in\mathcal{I}} e_{n(i)}^{(k)}e_1^{(d)\top},$$

where $R$ will be chosen later. Then, by approximate orthogonality,

$$v_{[\text{CLS}]}^\top W_Q W_K^\top X^\top \approx_{Rs\Delta} R\left(v_{[\text{CLS}]} + \sum_{i\in\mathcal{I}} b_i e_i^{(T)\top}\right),$$

so that by Lemma B.7,

$$\left\|\text{softmax}\left(v_{[\text{CLS}]}^\top W_Q W_K^\top X^\top\right) - \frac{1}{r+1}\left(e_{[\text{CLS}]}^{(T)\top} + \sum_{i\in\mathcal{I}} b_i e_i^{(T)\top}\right)\right\|_1 \le \frac{2T}{\exp(R - 2Rs\Delta)}.$$

Finally, we have

$$XW_V W_C e_1^{(d)} = -se_{[\text{CLS}]}^{(T)} + \sum_{i\in[T]} b_i \cdot v_i^\top v_\mathcal{I} \cdot e_i^{(T)} \approx_{s\Delta} -se_{[\text{CLS}]}^{(T)} + \sum_{i\in\mathcal{I}} b_i e_i^{(T)},$$

so that

$$|f_{\text{tf-head}}(X) \circ \Pi_1 - g_\mathcal{I}(r)| \le$$

$$\left\|\text{softmax}\left(v_{[\text{CLS}]}^\top W_Q W_K^\top X^\top\right) - \frac{1}{r+1}\left(e_{[\text{CLS}]}^{(T)\top} + \sum_{i\in\mathcal{I}} b_i e_i^{(T)\top}\right)\right\|_1 \left(\left\|XW_V W_C e_1^{(d)}\right\|_\infty + s\Delta\right)$$

$$+ \left\|\text{softmax}\left(v_{[\text{CLS}]}^\top W_Q W_K^\top X^\top\right)\right\|_1 (s\Delta)$$

$$\le \frac{2Ts(1+\Delta)}{\exp(R - 2Rs\Delta)} + s\Delta.$$

To get $(\gamma/4)$-uniform approximation, we choose

$$R = \frac{\log\left(\frac{8Ts(1+\Delta)}{\gamma - s\Delta}\right)}{1 - s\Delta}.$$

**Bag of vectors, $s$-dimensional output.** We use the same construction as above, except

$$W_C := \sum_{i\in\mathcal{I}} e_{n(i)}^{(k)}e_{n(i)}^{(d)\top}.$$

This will allow us to approximate the function

$$g_\mathcal{I}'(b) = \frac{1}{r+1}\sum_{i\in\mathcal{I}}(b_i - 1)e_{n(i)}^{(d)},$$

which is $(1/s)$-injective and has absolute value is bounded by 1. Then, for each $i \in \mathcal{I}$, we have

$$XW_V W_C e_i^{(d)} = -e_{[\text{CLS}]}^{(T)} + v_i^\top v_\mathcal{I} \cdot e_i^{(T)} \approx_{s\Delta} -e_{[\text{CLS}]}^{(T)} + b_i e_i^{(T)}.$$

Repeating the above analysis for each coordinate, we have

$$f_{\text{tf-head}}(X) \circ \Pi_s \approx_\varepsilon g_\mathcal{I}'(r),$$

where a slightly tighter bound

$$\varepsilon = \frac{2T(1+s\Delta)}{\exp(R - 2Rs\Delta)} + s\Delta$$

comes from the fact that $\left\|XW_V W_C e_i^{(d)}\right\|_\infty$ is now bounded by 1 instead of $s$. The previous choice of $R$ suffices for $(\gamma/4)$-uniform approximation. $\square$

*Proof of Lemma B.5.* This follows by instantiating Lemma B.8 with $\delta = \gamma/8, d_f = 1, n = s+1$. Notice that a $(\gamma/4)$-uniform approximation of a $\gamma$-strictly monotone function satisfies the conditions needed for Lemma B.8. $\square$

*Proof of Lemma B.6.* This follows by instantiating Lemma B.8 with $\delta = \gamma/8, d_f = s, n = 2^s$. Notice that a $(\gamma/4)$-uniform approximation of a $\gamma$-injective function satisfies the conditions needed for Lemma B.8. $\square$

## C   EXPERIMENT DETAILS

### C.1   EMPIRICAL SCALING LAWS FOR LEARNING SPARSE AND GATES

In this section, we provide details for the empirical sample complexity scaling law experiments, which are the main empirical verification of the $\log T$ dependence of the sample complexity arising from the analysis.

**Experimental setup.**   Synthetic tasks, parameterized by sample size $m$ and context $T$, were generated by the protocol described in the main paper: in each trial, one of the $\binom{T}{3}$ subsets of indices was selected uniformly at random, under i.i.d. random inputs $X \sim \mathrm{Unif}(\{0, 1\}^T)$. A 1-layer Transformer network (with the scalar output convention) was trained with Adam (Kingma & Ba, 2014) and batch gradients of the cross entropy loss for binary classification. For each choice of $T \in \{100, 150, \dots, 350, 400\} \cup \{500, 600, 700, 800\}$ and $B \in \{50, 60, 70, \dots, 200\}$, 50 independent trials were performed (re-randomizing the dataset generation, random initialization, and dropout masks). Cross-validation was performed on a holdout sample of size $10^4$ every 10 iterations. At the end of 1000 training iterations, the trial was counted as a success if the maximum validation accuracy throughout training was greater than 0.99. (In 100% of runs, the training loss was driven to $< 10^{-4}$, with 100% training accuracy, within 1000 iterations.)

**Architecture.**   Like (Chen et al., 2021a), our experimental setup is based on a popular PyTorch implementation (`https://github.com/karpathy/minGPT`), with some optimizations for faster 1-layer training and inference. This implementation includes widely-used architectural details which deviate slightly from the theoretical presentation; refer to the referenced repository for details. All hyperparameter settings left undiscussed are taken from the defaults in this codebase.

**Hyperparameters.**   A fixed architecture was used ($d = 64$, 16 parallel heads), with trainable positional embeddings initialized with Gaussian entries $\mathcal{N}(0, \sigma^2)$, $\sigma = 0.02$, 3 input token embeddings (corresponding to $0, 1, [\texttt{CLS}]$), and 2 output embeddings (corresponding to the possible labels $0, 1$). For regularization mechanisms, typical choices were used: 0.1 for {attention, embedding, output} dropout; $10^{-4}$ weight decay. The Adam optimizer was instantiated with typical parameters $\eta = 10^{-3}, \beta_1 = 0.9, \beta_2 = 0.999$.

**Results and discussion.**   Our findings are summarized by Figure 4, enlarged from the main paper. On the left, the fraction of successful trials is plotted for $T = \{100, 200, 400, 800\}$ with standard errors derived from the normal approximation. Notice that in this "low-data" regime, Transformer training is quite sensitive to the stochasticity in the experiments, including the training data sample (which needs to be sublinear in the context size), so there is no sharp "phase transition". On the right, critical sample sizes are shown, defined as the smallest $m$ for which the fraction of successes was greater than 0.5, with standard errors derived from 50 bootstrap samples. We observe that this architecture is able to solve this "planted sparse Boolean function" task with a sublinear scaling in the sample size.

**Infrastructure and computational costs.**   Each training run took at most 20 minutes on an NVIDIA RTX A6000 GPU (with most of computation time spent on cross-validation). Although these experiments are not in the same regime as state-of-the-art settings (such as BERT pretraining), they require optimized GPU implementations to run in a reasonable timeframe (days, as opposed to months).

### C.2   LEARNING PARITIES

A natural question for further exploration is whether Transformers can learn other Boolean functions. In particular, the statistical probe presented for the $s$-way AND function is equally valid for XOR. The latter is arguably more intriguing: parities are the basis elements in the monomial (i.e. Fourier) expansion of a Boolean function; (O'Donnell, 2021); furthermore, there are computational hardness barriers; see the works cited in the main paper for further context.

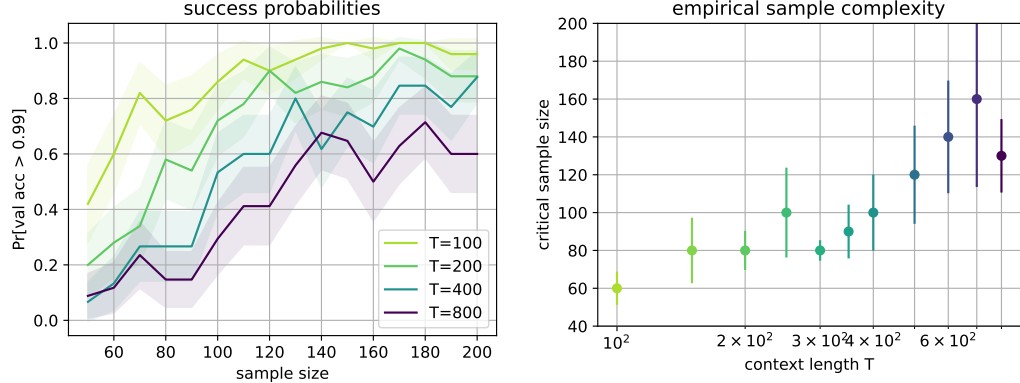

Figure 4: Enlarged plots from Figure 2, the main experimental validation of our theory.

**Experimental setup.**   These experiments match the setting of the main AND experiments, except for a few differences, which we enumerate here. Gradient-based training is done on streaming online losses (so that there is no training/validation split), with batch size 2048, for 10000 iterations (since parities take significantly longer to fit).

**Discussion.**   The key observation of this exploratory experiment is to point out the phenomenon that Transformer architectures can fit sparse parities at all (rather than to measure any particular trend), pointing to computational mysteries discussed in the main paper. We do not present empirical scaling laws for this set of experiments, as the experiments with small batch sizes are higher in variance, and we are unable to scale this phenomenon (i.e. learn any parities) at context sizes above $T = 50$.

## D   ADDITIONAL RELATED WORK

In this section, we discuss some additional related work.

**Attention-based architectures.**   Building upon the success of modern attention-based architectures, a large body of work (e.g. Goyal et al. (2020; 2021), and the works cited within) has sought to explicitly induce model sparsity and modularity in the architecture design. Our analysis is relevant to any architecture that uses a softmax (or similar) bottleneck for statistical capacity, and could inform design principles for norm-based capacity control of these architectures.

**Expressive power of Transformers.**   Several works establish results on the representational power of self-attention architectures in regimes where the statistical guarantees are necessarily weak or vacuous (i.e. there are too many functions in the class). Dehghani et al. (2018); Yun et al. (2019); Bhattamishra et al. (2020a;b) establish universal function approximation and Turing-completeness, which have been known for previous architectures (Siegelmann & Sontag, 1995). Our work is a significantly finer-grained analysis, in which we establish a hierarchy of function classes (indexed by sparsity $s$) representable by these architectures, with tight (in terms of $T$) statistical guarantees. Hron et al. (2020); Yang (2020) analyze properties of the kernels induced by Transformers at the infinite-width limit.

Likhosherstov et al. (2021) analyze the sparsity patterns representable by a self-attention head, with results superficially similar to ours: when the embedding dimension is at least logarithmic in the context length, all sparse matrices can be approximately realized by an attention head. However, their analysis is not about the capacity of the function class: it quantifies over the input $X$, and holds the parameters $(W_Q, W_K, \dots)$ to be constant (rather than vice versa). This finding serves as an interesting complement to our result: even though the attention mixture weights can take on exponentially many sparsity patterns for distinct inputs, the generalization error scales as $\log(T)$.

**Interpreting attention mixtures.** A line of empirical work ("BERTology") has made progress on understanding and interpreting state-of-the-art Transformer language models by examining the activations of their attention mechanisms (Clark et al., 2019; Tenney et al., 2019; Rogers et al., 2020). In some cases, these works have found instances in which Transformers seem to have learned features that are reminiscent of (sparse) hand-crafted features used in natural language processing, without explicit supervision. Our analysis formalizes the intuition that self-attention heads can represent sparse interactions within the context in a statistically meaningful way.

**Other theoretical work on self-attention.** Kerg et al. (2020) analyze self-attention and its benefit for learning long-term dependencies by establishing gradient norms bounds and showing how attention helps address the problem of gradient vanishing in recurrent networks. In contrast to our results that analyze the statistical and representational properties of attention-based architectures, this work focuses on the computational aspects of gradient-based methods on recurrent networks with self-attention.

**Synthetic experiments with Transformers.** Power et al. (2021) train small Transformer networks on synthetic algebraic tasks, and discover an abrupt phase transition from overfitting to correct generalization similar to ours. Tay et al. (2020) propose some synthetic tasks for benchmarking the ability of Transformer variants to capture long-range dependences. Chen et al. (2021a) present a synthetic demonstration of extrapolation (inferring a maximum-reward path from random walk observations) when using Transformers for offline reinforcement learning. Lu et al. (2021) probe the transfer learning capabilities of pretrained Transformers, and consider some simple Boolean tasks. Our experimental protocol of learning sparse Boolean functions provides a simple and fundamental setting for elucidating computational and statistical properties of sequence modeling architectures.

