# OpenReview forum: "Inductive Biases and Variable Creation in Self-Attention Mechanisms"
_ICLR.cc/2022/Conference — ICLR 2022 Submitted_

### Official Review · Reviewer_8Th6 · 2021-11-02

**Correctness:** 4
**Technical Novelty And Significance:** 4
**Empirical Novelty And Significance:** 3
**Recommendation:** 8
**Confidence:** 4

**Main Review:**

This paper adresses an important gap in theoretical understanding of attention-based systems that are widely used in state of the art models. It is very clearly written, despite the complexity of the subject matter and the difficulties inherent to the notation of attention methods. The motivation of the main findings is clear and timely, and I believe the general community will be interested and benefit from this contribution. Although I am not an expect in all the tools used in the derivations of the main results, I believe the work is correct, novel, and significantly contributes to our understanding of Transformer-like systems.

I believe there are two main points that could be improved to further strengthen this contribution.

1- Sparcity:
One of the central findings of this paper involves the sparsity of functions learned by attention systems, and more precisely attention heads as found in Transformers. Indeed, on page 6 immediately following Corollary 4.4, the authors write
"Our bounds have a logarithmic dependence on T highlighting the inductive bias of the transformer towards selecting sparse functions of the context"
I think the link between the bounded covering numbers and representation sparsity is not clearly fleshed out and could be unpacked a little further to appeal to a broader audience. Two things could be done.
First, for the interested non-expert, why does logarithmic dependence of bounds governing covering numbers lead to *sparse* functions ? Some context and details here would be welcome.
Second, most would already know informally that the softmax operation in attention weight computation leads to sparsity. This proof formalizes this in some way, and a discussion about the important role of the "Norm" component in the bounds, which is crucial, could be especially enlightening. Given these novel bounds, what would be the impact of using different Norm operations ? (e.g. Gumbel softmax v.s. softmax). Furthermore, not only is the softmax operation key in this result, it is also the major bottleneck for computational complexity, and a lot of work has been devoted to circumvent this problem (e.g. approximation of softmax by random kernels). How do the proposed bounds shed light on these issues ?

2- Experiments:
I understand that his is a theoretical contribution and that experiments play an supporting role in this paper. Nevertheless, the boolean functions experiments are Taylor-made to fit the bounds presented, and do not necessarily address the generalization properties associated with these bounds. A serious discussion about the type of experiments that could be carried out at scale by STOA systems, that would confirm the theoretical results, would strengthen the message. Moreover, two minor issues with regards to the experiment section remain. (1) While the finding of "sub-linear" scaling of sample complexity is consistent with scaling rates found analytically, the experimental results are far from confirming the same scaling. Sublinear can mean many things and is somewhat loose, and the experiments are noisy in that regard. A discussion about this limitation and what could be done to improve them would be welcome. (2) It appears this section might have been shortened or was added hastily as the main figure (Fig 2) has an incomplete caption (no description of the Left or center panel), and it appeals to the "{overfitting, correct} regimes" which are not discussed in the text (unless I missed it).

Some minor issues to correct:

* In definition 3.1, the Norm function definition involves a \Delta, I'm not sure what this represents. Is this a typo ? I might be mistaken but I don't recognize this as standard notation.

* In the proof overview of Thm 4, the second sentence is misswritten: "This property allows us to construct the cover by into covering..."

* The paragraph following Lemma 4.3 also suffers the same faith: "The key observation here is that the attention part of the network is computes using Norm..."

* In the description of the first numerical experiment in Section 5, it may be be good to explicitly refer to the appendix section containing more details as it is not clear how the task is implemented from the short description.

*In the Related work section, Theory for attention models, the authors might consider citing recent work directions establishing gradient norm bounds for (self-)attentive systems. e.g. "Untangling tradeoffs between recurrence and self-attention in artificial neural networks. Kerg et al., NeurIPS 2020"

**Summary Of The Paper:**

This paper provides rigorous norm-based bounds on the expressivity and inductive biases attention and self-attention provides when learning complex functions. The authors leverage established analytical tools involving complexity measurements, together with novel derivations for standard attention heads transformations,  to provide novel into generalization properties of Transformers, and into the types of functions they prefer to learn. Finally, the authors provide a set of simple numerical experiments using Boolean functions that serve as basic checks on their theory.

**Summary Of The Review:**

This paper is well-written, correct, and provides results that are significant and advance our understanding of attention mechanisms. I recommend publication. The reason I do not give it the highest score is because of two slight weaknesses: (1) lack of interpretation of the obtained bounds with respect to choices of Norm operation (2) Simplistic and only mildly convincing experiments.

---

> ### Author Response · Authors · 2021-11-22
> **Response to Reviewer 8Th6**
>
> We thank the reviewer for their support and detailed constructive feedback. Here we attempt to address the comments/questions raised by the reviewer:
>
> >  I think the link between the bounded covering numbers and representation sparsity is not clearly fleshed out and could be unpacked a little further to appeal to a broader audience. Two things could be done. First, for the interested non-expert, why does logarithmic dependence of bounds governing covering numbers lead to sparse functions ? Some context and details here would be welcome.
>
> Thank you for the suggestion. We have modified the introduction to be more explicit about the connection between our statistical and representational results.
>
> > Second, most would already know informally that the softmax operation in attention weight computation leads to sparsity. This proof formalizes this in some way, and a discussion about the important role of the "Norm" component in the bounds, which is crucial, could be especially enlightening. Given these novel bounds, what would be the impact of using different Norm operations ? (e.g. Gumbel softmax v.s. softmax). Furthermore, not only is the softmax operation key in this result, it is also the major bottleneck for computational complexity, and a lot of work has been devoted to circumvent this problem (e.g. approximation of softmax by random kernels). How do the proposed bounds shed light on these issues ?
>
> In our new version we have a more refined analysis for different Norm functions; the covering number bound now relies on an assumption about the Jacobian matrix of the Norm function (namely that the sum of the magnitudes of the entries of the Jacobian is bounded). We further show that softmax satisfies our assumption. We agree that analyzing different operations, such as the Gumbel softmax (which is probabilistic), is an interesting direction for future work, and the characterization in the new draft may help future work illuminate the tradeoff between computational complexity and usefulness for learning.
>
> > A serious discussion about the type of experiments that could be carried out at scale by STOA systems, that would confirm the theoretical results, would strengthen the message.
>
> For a discussion on why SOTA experiments are omitted, please refer to the general comment addressed to all reviewers.
>
> > While the finding of "sub-linear" scaling of sample complexity is consistent with scaling rates found analytically, the experimental results are far from confirming the same scaling. Sublinear can mean many things and is somewhat loose, and the experiments are noisy in that regard. A discussion about this limitation and what could be done to improve them would be welcome.
>
> Near the statistical limit of learning a $s$-sparse function (from $s \log T$ samples), Transformers exhibit unstable behavior: the solutions found by gradient descent depend heavily on the randomness of the sample as well as optimization (initialization, dropout, etc.). Reducing this instability is an important challenge in the practice of deploying Transformers. We will revise with more random seeds to tighten the error bars, and larger $T$ (not completed for this rebuttal period due to computation time limitations).
>
> > It appears this section might have been shortened or was added hastily as the main figure (Fig 2) has an incomplete caption (no description of the Left or center panel), and it appeals to the "{overfitting, correct} regimes" which are not discussed in the text (unless I missed it).
>
> We have fixed the figure caption in the updated version. {Overfitting, correct} refer to {m too small for the model to generalize, m sufficiently large}.
>
> > In definition 3.1, the Norm function definition involves a \Delta, I'm not sure what this represents. Is this a typo ? I might be mistaken but I don't recognize this as standard notation.
>
> $\Delta^{n-1}$ is used to denote the simplex in $n$ dimensions. We have added a definition in the notation section.
>
> > In the proof overview of Thm 4, the second sentence is miswritten: "This property allows us to construct the cover by into covering..."
> The paragraph following Lemma 4.3 also suffers the same faith: "The key observation here is that the attention part of the network is computes using Norm..."
>
> Fixed these.
>
> > In the description of the first numerical experiment in Section 5, it may be be good to explicitly refer to the appendix section containing more details as it is not clear how the task is implemented from the short description.
>
> Done.
>
> > In the Related work section, Theory for attention models, the authors might consider citing recent work directions establishing gradient norm bounds for (self-)attentive systems. e.g. "Untangling tradeoffs between recurrence and self-attention in artificial neural networks. Kerg et al., NeurIPS 2020"
>
> We have added a discussion of this work in Appendix D.

---

### Official Review · Reviewer_k53Q · 2021-11-02

**Correctness:** 3
**Technical Novelty And Significance:** 3
**Empirical Novelty And Significance:** 3
**Recommendation:** 6
**Confidence:** 3

**Main Review:**

Strengths:
1 This paper asks an essential question about the expressive power of attention and derives low-norm-based bounds for attention head, self-attention head, and the transformer model.

2 The bound in this paper has logarithm dependence on sequence length. These results indicate a finding that Transformer selects sparse functions of context and validates them with empirical analysis.

3 The theoretical analysis is rigorous with assumptions and detailed proofs.

Weaknesses:

1 The paper is not clear enough. The formulation is a little complicated, although the authors describe the well-known methods.
On page 5, the authors use the term tf-head before defining them; It would be better if the authors describe the two experimental tasks in detail;

2 This paper could do more practical experiments to show the usage of the bound or the finding of sub-linear scaling law.

3 The Related Work subsection has discussed many previous studies, but the paper does not compare or discuss previous methods in the theoretical analysis or in the empirical comparison.

Typos:
" (for eg, linear functions"-> e.g. linear functions, page 5



**Summary Of The Paper:**

This paper claim that they contribute to the norm-based generalization bound of the Transformer model. The paper uses its Lipschitzness to get the bound of self-attention model. With this bound, the paper analyses the inductive biases of self-attention modules; it investigates which function and the long-range dependencies the self-attention modules represent. This paper aims to answer this new question. It presents a theoretical finding that bounded-norm TransformerTransformer can fit sparse functions of the input sequence with sample complexity scaling only logarithmically with the context size. This paper also empirically validates this finding and the scaling law.


**Summary Of The Review:**

I like the contribution of generalization bounds for Attention and Transformer models. But there are some weaknesses. Therefore, my recommendation is marginal above the borderline.

---

> ### Author Response · Authors · 2021-11-22
> **Response to Reviewer k53Q**
>
> We thank the reviewer for their positive feedback. We address the major concerns listed by the reviewer.
>
> > The paper is not clear enough. The formulation is a little complicated, although the authors describe the well-known methods. On page 5, the authors use the term tf-head before defining them
>
> In the new version we have improved the presentation, making sure to define all functions and corresponding function classes (including tf-head) clearly.
>
> > It would be better if the authors describe the two experimental tasks in detail;
>
> The experimental tasks are described in detail in Appendix C, due to space considerations.
>
> > This paper could do more practical experiments to show the usage of the bound or the finding of sub-linear scaling law.
>
> For a discussion on why SOTA experiments are omitted, please refer to the general comment addressed to all reviewers.
>
> > The Related Work subsection has discussed many previous studies, but the paper does not compare or discuss previous methods in the theoretical analysis or in the empirical comparison.
>
> As described in our related work, various generalization bounds have been established for different architectures using covering-number arguments. These bounds do not extend directly to attention models due to the vast differences in the architectures of attention-models compared to MLPs and convolutional networks, which are more commonly studied. Our bounds also use covering number arguments but rely on a novel reduction to the $\ell_\infty$ covering number bound of [Zhang](https://www.jmlr.org/papers/volume2/zhang02b/zhang02b.pdf) for linear function classes, whereas other works that prove covering number bounds for neural networks reduce to $\ell_2$ covering number bounds.
>
> We have added a brief description of this in our related work section. As for empirical comparison, see our updated Appendix D.

---

> > ### Comment · Reviewer_k53Q · 2021-11-29
> > **Thank you for your response.**
> >
> > I appreciate your effort to address the concerns about clarity. However,  the other issues are not well addressed. So I recommend the weak acceptance.

---

### Official Review · Reviewer_YFzD · 2021-11-03

**Correctness:** 4
**Technical Novelty And Significance:** 4
**Empirical Novelty And Significance:** 2
**Recommendation:** 5
**Confidence:** 4

**Main Review:**

Pros:

- The paper is very well written.
- The paper tackles an important problem of studying the inductive bias of attention module.
- The idea of sparse variable creating is interesting and relevant to the larger deep learning community in general.

Cons:

- The experimental results are only on a synthetic task. It would be useful to study it on more complex tasks (as mentioned in the future work section of the paper).

Related Work:

- The basic idea involves the  sparsity of functions learned by attention systems, and more precisely attention heads as found in Transformers. The analogy to "attention heads" as functions in typed argument programming languages and representation of different "positions" as different variables has been made before such that different heads only need to attend to sparse subset of positions [1, 2].

- [1] RIMs, https://arxiv.org/abs/1909.10893
- [2] NPS, https://arxiv.org/abs/2103.01937

**Summary Of The Paper:**

The paper provides a theoretical analysis of the functioning of the self-attention modules. The paper shows that the Transformer architectures with some modifications (for ex. bounded norm) can create sparse variables. The paper proposes an experimental protocol to support the analysis.


**Summary Of The Review:**

The paper presents a theoretical analysis of the inductive bias of self-attention models, showing that the transformer architecture can learn sparse functions with some modifications.

---

> ### Author Response · Authors · 2021-11-22
> **Response to Reviewer YFzD**
>
> We thank the reviewer for their positive feedback. Given that their only concern appears to be our omission of experiments on non-synthetic data, we hope that the following explanation (copied from the response to all reviewers) addresses this concern and clarifies the scope/contribution of this paper. In light of this, we hope the reviewer will consider increasing their score.
>
> _Copied from the response to all reviewers on non-synthetic experiments:_
>
> - We reiterate that the primary scope of this paper is to provide theoretical foundations for analyzing the statistical and representational properties of self-attention. Our work complements experimental findings found in the SOTA NLP literature:
>     - BERTology: the “sparse variable creation” inductive bias formalizes the widespread finding that attention heads learn features that correspond to (sparse) hand-crafted features used in NLP (dependencies, coreferences, ..). We have added a discussion of these works in Appendix D.
>     - Practitioners vary the context length $T$ in practice without needing to impose significant additional regularization controls: e.g. Transformer-XL [Dai et al. ‘19], Linformer [Wang et al. ‘20]. The practical implication of our main result is not that it predicts that the generalization error will scale logarithmically-- it is a generalization upper bound which explains why the capacity scales only mildly with $T$ (unlike $\ell_2$ regression or $T$-gram models, which require aggressive regularization to handle rich contexts).
> - We agree that SOTA NLP experiments to probe sparsity and the sample complexity scaling laws are worthy of follow-up work. Formulating the exact experiments here is not so straightforward, due to several confounding factors that could impact the sample complexity.
>     - A fully non-synthetic experiment setup would entail fixing the task (e.g. next-word prediction), varying the context length $T$, and investigating the effect of $T$ on generalization error. Here, a significant confound presents itself: the length of the context alters the objective and complexity of the task. Ignoring this confound, there is plenty of evidence (e.g. the comparisons in Transformer-XL & Linformer) that scaling up $T$ improves end-to-end performance on many tasks without significantly degrading generalization.
>     - An alternative would be a semi-synthetic experiment carefully designed such that increasing the context length doesn’t provide more information towards solving the task. For example, one could consider adding irrelevant text to a passage in a question-answering task, or irrelevant facts to the premise of an entailment task. Creating a procedure to generate these augmented datasets while preserving the “naturalness” of the distribution and without creating contradictions is a challenging problem in itself, and certainly beyond the scope of this paper.
> - Because we weren’t fully satisfied with any of the above approaches towards probing the scaling laws, we designed a synthetic experiment based on the classical setup of sparse boolean functions, where we could precisely control the statistical limit (i.e. $\Omega(s \log T)$ samples) and isolate the problem of identifying the relevant parts of the context. We believe our proposed setup stands independently as a useful benchmark for the capacity control of self-attention networks. We hope that the discussion on related synthetic experiments added to Appendix D clarifies this.
>
> > Related work
>
> Thank you for pointing us to these relevant papers. We have included them in our updated version (Appendix D).

---

> > ### Comment · Reviewer_YFzD · 2021-11-29
> > **Thanks for explaining.**
> >
> > "We reiterate that the primary scope of this paper is to provide theoretical foundations for analyzing the statistical and representational properties of self-attention. Our work complements experimental findings found in the SOTA NLP literature"
> >
> > This is useful for me.
> >
> > "An alternative would be a semi-synthetic experiment carefully designed such that increasing the context length doesn’t provide more information towards solving the task. For example, one could consider adding irrelevant text to a passage in a question-answering task, or irrelevant facts to the premise of an entailment task. Creating a procedure to generate these augmented datasets while preserving the “naturalness” of the distribution and without creating contradictions is a challenging problem in itself, and certainly beyond the scope of this paper."
> >
> > This is an interesting idea. One simple way to implement it would be a "copy" task, where their are some bits of information (information phase), and then distractor phase, and then after an event signal, the model is asked to copy the bits present in the information phase. One way to analyze it would be by varying the  number of bits in the distractor phase (irrelevant information). Normally transformers struggle on such tasks. A more realistic version could be by varying the resolution of MNIST digit.

---

> > > ### Author Response · Authors · 2021-11-29
> > > **Follow-up re: semi-synthetic experiments**
> > >
> > > Thanks for the thoughtful response.
> > >
> > > - Note that the proposed "information + distractor" setup is, in fact, very close to the setup of our synthetic experiments-- the only difference is that the sparse "copy" operation is replaced with a Boolean function (which can be thought of as a hash of the relevant bits, requiring the model to identify them). We decided that it would be clearer to present this as a synthetic experiment, rather than add superficial elements to turn this into a semi-synthetic NLP setting. We expect the same results to hold for the "copy" variant, and will investigate.
> > > - We are not sure how to interpret "varying the resolution of MNIST digits". Possible interpretations:
> > >     - If you meant "increase the image dimensions, padding with irrelevant pixels": the same discussion as above would hold. The additional sample complexity of learning the image features would be a significant confounding factor towards measuring empirical scaling laws. Evidence that Transformers can solve visual tasks with distractions (at all, without measuring sample efficiency) is present in "Long Range Arena: A Benchmark for Efficient Transformers", [Tay et al. '20].
> > >     - If you meant "upsampling/downsampling the MNIST images": this is an interesting suggestion-- varying the amount of *redundant* rather than *distracting* context. One major confounding factor with this methodology: given that applying Transformers in vision seems to require fixed-size patch embeddings at the first layer, the function class favors a particular scale. Thus, changing the resolution while holding the architecture constant changes the task. Even without fixed patch sizes, it is unclear how to keep the learned features constant while varying the scale of the input, since the sparsity of these features will change with the scale.
> > >
> > > We hope that your concerns have been addressed, and that this reflects in your updated score.

---

### Official Review · Reviewer_yc2n · 2021-11-03

**Correctness:** 3
**Technical Novelty And Significance:** 3
**Empirical Novelty And Significance:** 3
**Recommendation:** 3
**Confidence:** 3

**Main Review:**

1. The writing might need some improvements. The notation m and T were initially very hard to understand, until to the final page, I realize m denotes sample size while T denotes context length.

2. At Corollary 4.4, the paper mentioned "...highlighting the inductive bias of the transformer towards selecting sparse functions of the context" and then later in section 5, "The theory predicts the sample complexity of learning a bounded-norm transformer should match this scaling in T". This is very interesting to me, but not sure how it is verified in Fig 2. There are some log-scaled correlation in the middle figure, but don't know what it says. For instance, the sample complexity should match the scaling of T, but for what goal?

3. Based on the main takeaway I learnt from this paper, I wonder if it means, imagine in a coreference resolution task, the mention coreference is super dense while the context length is short. This could be challenging for a transformer to handle. But what if keeping the same amount of coreference links but synthetically increase the context size, would the task become easier? I imagine this can be a more real task (even though it might still be synthetic, but more realistic than a dataset of boolean functions) to experiment with. Furthermore, having a realistic data is good to show if the main conclusion of this paper is valid and the bound is a tight one in real application. Otherwise, the logarithmic dependency might not be established.


**Summary Of The Paper:**

This paper proposes a theoretical bound on inductive biases self-attention module can represent. It gives detailed walkthrough about how such bound can be derived and the result complements conclusion from prior works. The conclusion of this paper is very interesting that the complexity of sparse boolean function that a self-attention module can capture is logarithmically dependent of the context size. It could cast insights on future probing works and dataset designs. A downside, however, is the experiment part which only focused on a synthetic task. Analysis is largely absent.


**Summary Of The Review:**

I think this paper proposes an interesting conclusion from its derivation of the theoretical bound on self-attention modules. The amount of contribution on the theoretical part seems substantial. But the verification part is weak. I don't feel I am confident to eat the takeaway from what I see in the experiment section.

---

> ### Author Response · Authors · 2021-11-22
> **Response to Reviewer yc2n**
>
> We thank the reviewer for their useful feedback. Here we address the major concerns raised by the reviewer:
> > “ The notation m and T ... m denotes sample size while T denotes context length.”
>
> Yes, that is the correct interpretation of $m$ and $T$. We have made the distinction between $m$ and $T$ explicit at the beginning of Section 2.
>
> > Explaining Fig 2.: “the sample complexity should match the scaling of T, but for what goal?”
>
> The goal of the experiment presented in Figure 2 was to verify the logarithmic scaling law for sparse functions suggested by the theory. In particular, Corollary 4.4 states that the capacity (log covering number) of the class of bounded-norm Transformer self-attention heads grows only logarithmically with T, the context length. This implies that the sample size needed to achieve a small generalization gap on any task also provably grows only logarithmically with T, as long as the weight norms are bounded. See Lemma 2.2 in the updated draft for a precise generalization bound.
>
> Proposition 4.7 in our updated draft (corresponding to Propositions 4.7 and 4.8 in the submission) states that sparse functions can be represented by bounded-norm Transformer heads. Thus, a behavior we might expect is for a Transformer head to actually fit such a sparse function without too much blow-up in the norms, and by the capacity results, this would guarantee that the number of samples required to generalize should grow only logarithmically in the context length.
>
> Now look at the center plot of Figure 2—it shows that when a Transformer head is trained on a particular family of sparse functions (sparse conjunctions), it requires roughly $O(\log(T))$ samples in order to generalize to unseen data (the x axis is logarithmically scaled). Which is what the theory predicts! (As the right-hand figure shows, even with fewer samples the network still achieves zero training error, so the generalization gap is identical to the test error. Appendix C has more details about the experiment.
>
> > “But what if keeping the same amount of coreference links but synthetically increase the context size, would the task become easier?”
>
> Synthetically increasing context size will not make the task easier since we still need to identify the same coreference links. However, our results do point out that adding synthetic context size (or having part of the context that is irrelevant to the task) does not make the task much harder in terms of sample complexity. This is because our bounds scale only logarithmically with the context length.
>
> > Furthermore, having a realistic data is good to show if the main conclusion of this paper is valid and the bound is a tight one in real application. Otherwise, the logarithmic dependency might not be established.
>
> For a discussion on why SOTA experiments are omitted, please refer to the general comment addressed to all reviewers. We stress that the practical implication of our main result (e.g. for large-scale language modeling) is not that it predicts that the generalization error will scale logarithmically-- it is a generalization upper bound which explains why the capacity scales only mildly with $T$ (unlike $\ell_2$ regression or $T$-gram models, which require aggressive regularization to handle rich contexts).

---

### Author Response · Authors · 2021-11-22
**Manuscript update and general response to all reviewers**

We thank all the reviewers for their thoughtful feedback. We have uploaded a revision of the manuscript, with changes listed below:

- A more general analysis of the normalization function, to handle cases other than softmax (Assumption 4.1, (3)).
- In the function approximation results (Section 4 & Appendix B), alternative setups (trainable & untrainable position embeddings) to accompany the original permutation-invariant “bag-of-vectors” construction.
- A more explicit review in the main paper of how covering number bounds imply generalization bounds (Lemma 2.2).
- Clearer definitions and corrected typos.
- Discussion in the Intro of the connection between the generalization results and the (sparse) representation results
- Additional related work (Appendix D).

**Discussion on non-synthetic experiments:** All of the reviewers highlighted the fact that our experiments are only on synthetic data.

- We reiterate that the primary scope of this paper is to provide theoretical foundations for analyzing the statistical and representational properties of self-attention. Our work complements experimental findings found in the SOTA NLP literature:
    - BERTology: the “sparse variable creation” inductive bias formalizes the widespread finding that attention heads learn features that correspond to (sparse) hand-crafted features used in NLP (dependencies, coreferences, ..). We have added a discussion of these works in Appendix D.
    - Practitioners vary the context length $T$ in practice without needing to impose significant additional regularization controls: e.g. Transformer-XL [Dai et al. ‘19], Linformer [Wang et al. ‘20]. The practical implication of our main result is not that it predicts that the generalization error will scale logarithmically-- it is a generalization upper bound which explains why the capacity scales only mildly with $T$ (unlike $\ell_2$ regression or $T$-gram models, which require aggressive regularization to handle rich contexts).

- We agree that SOTA NLP experiments to probe sparsity and the sample complexity scaling laws are worthy of follow-up work. Formulating the exact experiments here is not so straightforward, due to several confounding factors that could impact the sample complexity.
    - A fully non-synthetic experiment setup would entail fixing the task (e.g. next-word prediction), varying the context length $T$, and investigating the effect of $T$ on generalization error. Here, a significant confound presents itself: the length of the context alters the objective and complexity of the task. Ignoring this confound, there is plenty of evidence (e.g. the comparisons in Transformer-XL & Linformer) that scaling up $T$ improves end-to-end performance on many tasks without significantly degrading generalization.
    - An alternative would be a semi-synthetic experiment carefully designed such that increasing the context length doesn’t provide more information towards solving the task. For example, one could consider adding irrelevant text to a passage in a question-answering task, or irrelevant facts to the premise of an entailment task. Creating a procedure to generate these augmented datasets while preserving the “naturalness” of the distribution and without creating contradictions is a challenging problem in itself, and certainly beyond the scope of this paper.
-Because we weren’t fully satisfied with any of the above approaches towards probing the scaling laws, we designed a synthetic experiment based on the classical setup of sparse boolean functions, where we could precisely control the statistical limit (i.e. $\Omega(s \log T)$ samples) and isolate the problem of identifying the relevant parts of the context. We believe our proposed setup stands independently as a useful benchmark for the capacity control of self-attention networks. We hope that the discussion on related synthetic experiments added to Appendix D clarifies this.

---

### Decision · Program_Chairs · 2022-01-20

**Decision:**

Reject

**Comment:**

This paper presents a theoretical analysis of self-attention modules, using Lipschitz conditions.

It suffers from two main weaknesses: the clarity of the presentation, and the weak experimental section.